# Hardening in Au-Ag nanoboxes from stacking fault-dislocation interactions

Radhika P. Patil [1], David Doan[1], Zachary H. Aitken[2], Shuai Chen [2], Mehrdad T. Kiani[3], Christopher M. Barr[4], Khalid Hattar[4], Yong-Wei Zhang[2] & X. Wendy Gu [1✉]

Porous, nano-architected metals with dimensions down to ~10 nm are predicted to have extraordinarily high strength and stiffness per weight, but have been challenging to fabricate and test experimentally. Here, we use colloidal synthesis to make ~140 nm length and ~15 nm wall thickness hollow Au-Ag nanoboxes with smooth and rough surfaces. In situ scanning electron microscope and transmission electron microscope testing of the smooth and rough nanoboxes show them to yield at 130 ± 45 MPa and 96 ± 31 MPa respectively, with significant strain hardening. A higher strain hardening rate is seen in rough nanoboxes than smooth nanoboxes. Finite element modeling is used to show that the structure of the nanoboxes is not responsible for the hardening behavior suggesting that material mechanisms are the source of observed hardening. Molecular dynamics simulations indicate that hardening is a result of interactions between dislocations and the associated increase in dislocation density.

[1] Department of Mechanical Engineering, Stanford University, Stanford, CA 94305, USA. [2] Institute of High Performance Computing, A*STAR, 1 Fusionopolis Way, #16-16 Connexis, Singapore 138632, Singapore. [3] Department of Materials Science and Engineering, Stanford University, Stanford, CA 94305, USA. [4] Materials, Physical, and Chemical Sciences, Sandia National Laboratories, Albuquerque, NM 87185, USA. ✉email: xwgu@stanford.edu

Porous, nano-architected metals are of interest as light-weight, multifunctional structural materials[1–3]. A major benefit of these nanostructured metals is the ability to use nanoscale size effects to enhance strength and control deformation[4,5]. For instance, metallic nanowires can have extraordinary strengths of >6 GPa[5,6]. Plasticity within these nanostructures is governed by dislocation nucleation rather than the interactions between dislocations and other microstructural features as seen in bulk metals, which leads to novel deformation mechanisms such as shape memory effects[7]. These size-dependent properties can be combined with a 3D structural architecture that has an optimized mechanical response to achieve high strength and stiffness, or other properties such as a negative Poisson's ratio. This has been demonstrated using computational modeling[8,9], but has not been achieved experimentally. For instance, molecular dynamics simulations show that architected octet nano-lattices with single-crystalline Cu struts of ~5 nm in size should have a yield strength of ~500 MPa, but these structures cannot yet be fabricated[8].

Size effects are significantly important as dimensions approach ~10 nm scale. However, it remains challenging to fabricate architected metals with dimensions down to this order using current fabrication methods. Porous metals with ordered structural architecture have been fabricated by using polymer templates and subsequent metal deposition using electroplating or sputtering, but the resulting structures have metal features of ~100 nm to microns in size[1,10–12]. Metal 3D printing is another option for the fabrication of architected metals, but current methods are limited to ~150 nm features, and restricted to simple geometries without significant overhangs[13–15]. Disordered metallic nano foams are made by electrochemically dealloying a bimetallic alloy, which results in disordered microstructures with lower specific strength and stiffness than ordered architectures. Nanoscale size effects have previously been observed in such nano-foams with ligament sizes of 5–100 nm[16–19]. These foams have yield strengths of up to ~170 MPa, which indicates that ligaments have strength of ~4.6 GPa. The ability to create ordered structures at similar length scales should result in further improvements over such disordered structures.

Bottom-up colloidal synthesis is a method for making nano-architected metals that have been widely used in optics, catalysis, and electronics[20–22]. Solid metallic nanocrystals can be synthesized with a range of shapes (e.g. spheres, cubes, bipyramids, and octahedral) and sizes of 100 nm[23,24]. These nanocrystals can be transformed into porous nano-architected structures with wall thicknesses down to 5 nm. For instance, nanoboxes (hollow interior, solid faces) and nanocages (hollow interior, open faces) can be formed from solid cubes. In this way, these colloidal metals combine the ordered structural architecture and nanoscale dimensions that have been challenging to achieve in top-down structures. It is of interest to determine whether nanoscale size effects can be observed in these nano-architected metals or if novel deformation mechanisms take place. Hollow spheres made of CdS, carbon, and silica have been found to have high compressive strength[25–27], but colloidal metallic nanostructures with complex geometries have not yet been studied.

Here, we synthesize ~120–140 nm hollow, cubic Au–Ag nanoboxes with wall thickness of ~15 nm and determine their mechanical response using in situ scanning electron microscope (SEM) and in situ transmission electron microscope (TEM) compression testing. Smooth and rough nanoboxes are tested to evaluate the influence of surface roughness on strength and deformation. Smooth nanoboxes are found to have a yield strength of 130 ± 45 MPa, while rough nanoboxes have a yield strength of 96 ± 31 MPa. Both types of nanoboxes show significant hardening after yield, with rough nanoboxes having a

higher hardening rate. Finite element method (FEM) analysis is used to determine the influence of structural architecture on nanobox deformation and molecular dynamics (MD) simulations are used to understand the atomistic deformation mechanisms responsible for the hardening behavior. The simulations show that the observed strain hardening is not a result of the structure and geometry of the nanobox, rather, it results from the material's response involving accumulation of crystalline defects such as dislocations and stacking faults within the walls of the nanoboxes.

## Results

**Synthesis and characterization of Au–Ag nanoboxes.** Solid Ag nanocubes with {100} surfaces are synthesized using the polyol-based method[28]. Hollow nanoboxes are fabricated through galvanic replacement of the solid Ag nanocubes[21,29]. In the galvanic replacement reaction, $HAuCl_4$ is added to a solution of Ag nanocubes. $HAuCl_4$ dissociates into $H^+$ and $AuCl_4^-$ and reacts with the Ag nanocubes in the following reaction[30]:

$$3Ag + AuCl_4^- \rightarrow Au + 3Ag^+ + 4Cl^- \qquad (1)$$

This reaction results in the removal of Ag and the deposition of Au. Hollow nanoboxes are obtained from this reaction by Au depositing epitaxially on the surfaces of the Ag nanocube, which blocks further removal of Ag from that region. After this, the removal of Ag occurs at holes that are formed on the surface of the nanocube. Electrons generated during this galvanic reaction travel to the surface of the nanocube and more Au atoms are deposited on the surface[20,31]. The Ag nanocubes used in these experiments are 103 ± 8 nm in length, having rounded corners with radius of curvature of 11 ± 2 nm. A low concentration of $HAuCl_4$ results in smooth nanoboxes, while a higher concentration results in rough nanoboxes (Fig. 1). The smooth and rough nanoboxes have sizes of 139 ± 9 nm and 121 ± 12 nm, respectively (Fig. 2). Both smooth and rough nanoboxes contain holes at the

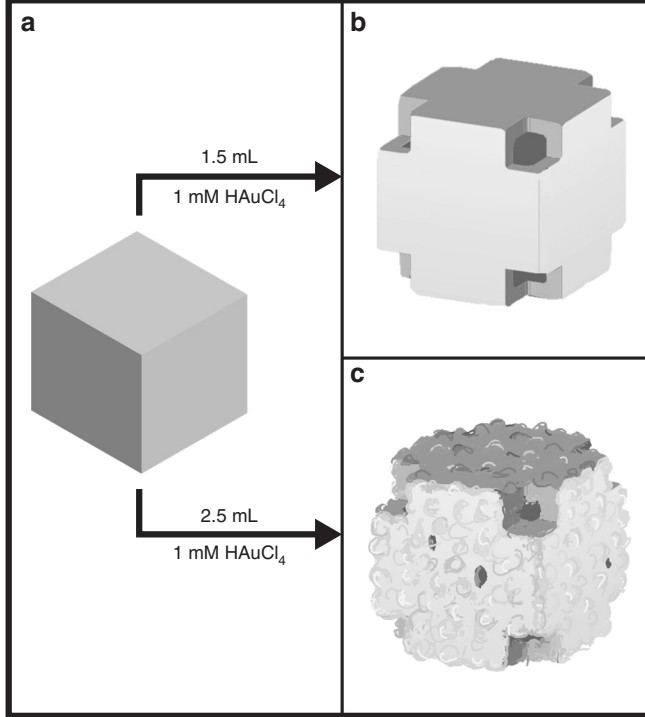

**Fig. 1 Schematic of nanobox synthesis. a** Galvanic replacement of a solid Ag nanocube results in **b** a smooth Au–Ag hollow nanobox when a low concentration of $HAuCl_4$ is used, and **c** a rough Au–Ag hollow nanobox when a low concentration of $HAuCl_4$ is used.

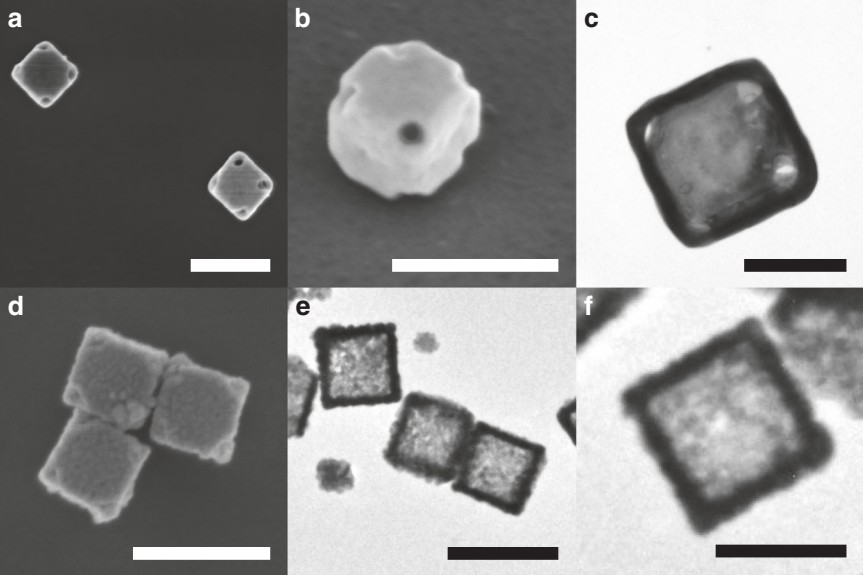

**Fig. 2 SEM and TEM images of Au–Ag nanoboxes. a–c** smooth and **d–f** rough nanoboxes. Scale bar is 200 nm in **a**, **b**, **d**, **e** and 100 nm in **c**, **f**. **b** was taken at a tilt angle 45°.

corners of the cubes, which has also been observed in previous studies on nanobox[20]. A nanocube with sharp corners has only {100} facets exposed but a nanocube with rounded corners has some {111} character at the corners. The {111} facets are poorly passivated in comparison to {100} facets, which results in the preferential removal of Ag atoms at the corners to create pits that grow into holes[20,22,30]. The holes at the cube corners have a diameter of $19 \pm 4$ nm in both the smooth and rough nanoboxes. TEM images show that the smooth and rough nanoboxes have wall thicknesses of $18 \pm 3$ nm and $14 \pm 4$ nm, respectively (Fig. 2). Each rough nanobox also contains 4–5 holes of characteristic diameter of $11 \pm 7$ nm randomly distributed on the walls, as measured using SEM images. The surface protrusions on the rough nanoboxes are estimated to have a height $10 \pm 3$ nm using TEM. The nanoboxes studied here are expected to be single crystalline. Previous studies, that used identical synthetic protocols, have used high-resolution TEM imaging and TEM diffraction to show that hollow structures formed from Ag nanocubes using galvanic exchange are single crystalline[32,33].

The resulting nanoboxes contain both Au and Ag, with the rough nanoboxes having a higher ratio of Au:Ag than the smooth nanoboxes. According to the galvanic exchange reaction (Eq. (1)), three Ag atoms are replaced by one Au atom, which should lead to an overall decrease in solid volume (volume of the nanobox excluding the hollow interior). By comparing the initial volume of the solid Ag nanocube, and the final volumes of the smooth and rough nanoboxes, we determine that the galvanic exchange reaction does not go to completion. For instance, the solid volume of the smooth nanoboxes is higher than that of the solid Ag nanocube, which indicates that the smooth nanoboxes contain both Au and Ag. The rough nanoboxes have ~17% lower volume than the solid Ag nanocubes, which indicates that the rough nanoboxes contain a higher ratio of Au:Ag than the smooth nanoboxes. This analysis is in agreement with previous energy dispersive spectroscopy (EDS) of hollow nanocubes and rough nanocages, which showed an Au:Ag ratio of 30:70 and 70:30, respectively[34,35]. This difference in composition is not expected to cause a significant difference in the properties of the smooth and rough nanoboxes. The Ag and Au interface does not impede the passage of dislocations or stacking faults, so the difference in

composition should not affect the deformation mechanism for the nanoboxes[36].

**In situ testing of nanoboxes.** The nanoboxes were compressed using in situ SEM mechanical testing. Nanoboxes are imaged before and after compression testing to determine the change in dimensions during the test. Nanoboxes are deposited onto a Si wafer and compressed using a diamond flat punch in SEM. The structures are compressed at a constant loading rate of $1\ \mu Ns^{-1}$ for all samples, which is equivalent to a strain rate of $\sim 0.01$–$0.02\ s^{-1}$, with a 50% strain stopping condition. Ten and eleven compression tests are performed on the smooth and rough nanoboxes, respectively. The compressive engineering stress-strain curves for both samples show an initial linear loading region followed by yielding (Fig. 3). We define the yield point as the limit of the linear region of the stress-strain plot. Yielding occurs at $130 \pm 45$ MPa for the smooth nanoboxes and $96 \pm 31$ MPa for rough nanoboxes. In the compressive stress-strain curves for the smooth nanocubes, the initial region is a linear regime up to the yield point. A range of behaviors is observed in the smooth nanoboxes after the yield point. Two samples show a load drop followed by hardening (blue curve in Fig. 3a). Another sample shows a plateau in stress and then hardening (red curve in Fig. 3a). The remaining samples show hardening after yielding, and increased hardening rate after 60% strain (black and pink curves in Fig. 3a). In contrast, the majority of the rough nanoboxes show continuous hardening (Fig. 3b). A small plateau after yielding is observed in a few of the samples (pink curve in Fig. 3b). Despite differences in the shapes of the stress-strain curves, the stress is ~250 MPa at 60% strain for both the smooth and rough nanoboxes, suggesting that the structural influence on deformation is reduced as the porous structure densifies. The hardening rate of the nanoboxes is quantified using Eq. (2)[37]:

$$\sigma_{pl} = K\varepsilon^{n} \qquad (2)$$

Here, K is the strength coefficient, $\sigma_{pl}$ is the plastic stress, $\varepsilon$ is the corresponding total strain, and $n$ is the strain hardening exponent. The strain hardening exponent is found to be 0.3 for the smooth nanoboxes and 0.7 for the rough nanoboxes. This indicates that

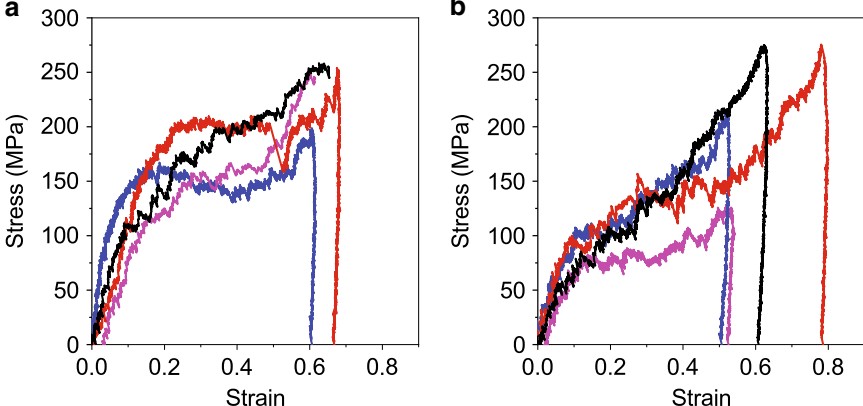

**Fig. 3 Representative engineering stress-strain plots for in situ SEM compression of Au–Ag nanoboxes. a** Smooth and **b** rough nanoboxes.

the rough nanoboxes have a higher hardening rate than the smooth nanoboxes.

In summary, the experimental stress-strain curves for the rough nanoboxes show a lower yield strength and a greater degree of hardening than the smooth nanoboxes. Although the dimensions of the walls differ between the smooth and the rough nanoboxes, the difference in yield strengths cannot be attributed to nanoscale size effects. The characteristic length scale of interest in the nanobox is the thickness of the sidewall, as this is the load-bearing structure within the nanobox. The rough nanoboxes have thinner walls than the smooth nanoboxes. Considering the "smaller is stronger" size effect in nanoscale single-crystalline metals[38], the thinner walls should be stronger than the thicker ones, which is not observed. Other researchers have proposed a size-independent strength when metallic nanostructures are in the 200 nm regime[39]. This would result in rough nanoboxes with the same strength as smooth nanoboxes per nanobox cross-sectional (load-bearing) area, which is also not observed. We postulate that the rough protrusions act as stress concentrations that reduce the barrier for surface defect nucleation. This is explored further using MD simulations and analytical analysis in the following sections.

The mechanical response of the hollow nanoboxes is substantially different from that of solid, single-crystalline metallic nanostructures, but is similar to nano- and micro-porous metal foams, and thin-shelled nanostructures such as carbon nanotubes[17,40–42]. Single crystalline metallic nanopillars and nanowires tested in compression show a high yield strength of up to 1.5 GPa, jerky plastic flow indicative of dislocation avalanches and no hardening[43–45]. This behavior has been shown in the solid Ag nanocubes that are used to form the nanoboxes[46]. In contrast, the hollow nanoboxes deform gradually, without large slip events, and show significant strain hardening. Metallic foams and thin-shelled nanostructures also have compressive stress-strain curves with gradual deformation, and hardening. This difference is primarily due to deformation modes such bending and buckling, which are not present in solid single-crystalline nanostructures under uniaxial compression. For instance, nanoporous gold nanowires of diameter 30–70 nm with ligament sizes 5–10 nm yield at ~100 MPa stress, and show considerable strain hardening beyond the yield point[17]. Micro-porous open-cell metallic foams show a stress drop after yielding, which is followed by strain hardening[40,47]. This stress-strain response is also observed in individual carbon nanotubes loaded in compression that deform through shell buckling[42].

A post compression SEM image of a smooth nanobox shows that it is 32% wider after compression (Fig. 4a), which is larger than the shape change that is expected from elastic deformation

alone. If the load-bearing walls of the nanoboxes were loaded in uniaxial compression up to the yield point and had the properties of solid Ag nanocubes (yield strength of 637 ± 251 MPa), an estimated nominal yield stress of ~270 MPa is required for the deformation[46]. This indicates that the nanobox has deformed plastically by a mode other than uniaxial compression. The compressive yield strength of the load-bearing walls maybe even higher than that of solid Ag nanocubes due to the small width of the walls. For instance, Ag nanowires with diameters of ~40 nm have been observed to have yield strengths of ~4.5 GPa[6]. If the walls of the nanoboxes have this high strength, then the nanoboxes can be expected to have a yield strength of ~2 GPa in uniaxial compression. The measured nanobox yield stress of ~130 MPa is much lower than these estimates, which indicates that another deformation mode takes place.

These results can be compared with in situ TEM compression tests on the nanoboxes, which were performed at a strain rate of ~0.03 s$^{-1}$. The in situ TEM movie of nanobox compression (Supplementary Movie 1) shows that the nanobox sidewalls appear to thicken after the nanobox makes contact with the compression plate, and the holes at the corners of the nanobox become elongated. Lines of dark and bright contrast can be observed on the front/back facing sidewalls, which may correspond to the formation of stacking faults. Comparison of pre- and post-compression TEM images that were extracted from the in situ TEM movie (Fig. 4 b, c) show a ~10% increase in the width of the nanobox after compression, and no evidence of bending in the sidewalls. Differences in the post compression in situ SEM and TEM mechanical testing images may be due to differences in strain rate, noise levels, or subtle differences in nanobox geometry. In addition, the experimental stress-strain curves show a range of behaviors for both the smooth and rough nanoboxes, which may result in different deformation modes.

**Finite element modeling.** For further insight into the influence of structural architecture on deformation, a hollow smooth nanobox is simulated using FEM analysis. The purpose of this FEM simulation is to determine the structural response for a material that does not include strain hardening. This is relevant because the solid Ag nanocube (the parent structure from which the nanobox is formed) does not show strain hardening under compression[46]. In this way, the FEM simulation reveals the deformation of a hollow box, without consideration of material or size effects. Thus, an elastic-plastic material model that does not include strain hardening was used in the FEM simulation, to mimic the material response of a Ag single-crystal nanocube[46]. The FEM model was then constructed to have a similar geometry

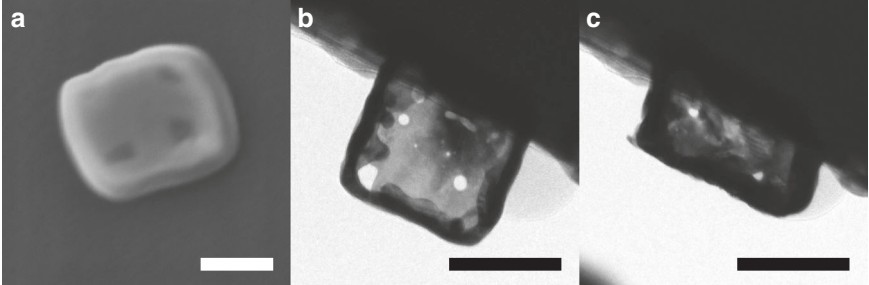

**Fig. 4 Images of deformed nanoboxes. a** Post compression SEM image at a 40° tilt. **b** Pre and **c** post compression TEM images during in situ TEM compression testing. Scale bar is 100 nm in all images.

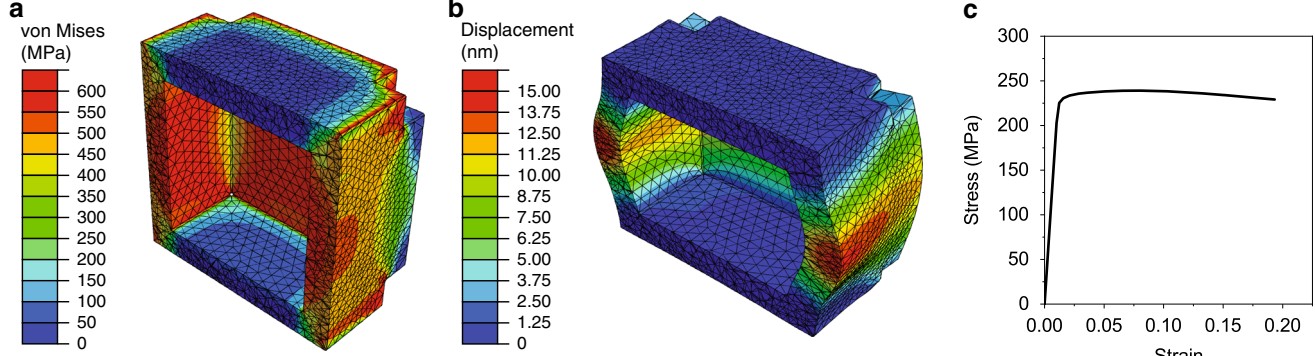

**Fig. 5 FEM simulation of a smooth nanobox.** Cross-sectional view of **a** von Mises stress at ~1.5% strain, **b** radial displacement at ~18% strain, and **c** corresponding engineering stress-strain curve.

as the experimental nanobox, with smooth walls and a hollow interior. The simulated nanobox is deformed in compression, resulting in compression and bending of the sidewalls. The simulations show high-stress concentrations at the corners of the nanobox and the sidewalls due to bending, relative to the rest of the nanobox (Fig. 5a). The bending of the sidewalls also contributes to a significant increase in the width of the nanobox. At ~20% strain, the width of the nanobox increases by ~20% (Fig. 5b). The resulting force-displacement curve is converted into an engineering stress-strain curve by normalizing to the initial footprint of the nanobox (Fig. 5c). This stress-strain curve shows a linear elastic regime that is followed by yielding at ~230 MPa. After yield, there is a small increase in stress followed by a decrease in stress after 8% strain. This curve shape is also observed in the uniaxial compression of cylindrical shells, with variations of the curve shape-changing depending on the wall thickness to size ratio[48,49]. While the structural changes in the simulated nanobox are qualitatively similar to the experimental nanoboxes (Fig. 4a), FEM analysis does not capture the increase in stress that is observed in both the smooth and rough nanoboxes after yield. This indicates that the hardening behavior observed in the experimental nanoboxes results from material rather than structural effects.

**Molecular dynamic simulations**. The influence of material properties and size effects on deformation mechanisms are investigated using MD simulations. Ag nanoboxes were simulated with a wall thickness to length ratios of 1:7 (similar geometry as the experimental nanoboxes), and an overall size of 50 nm. They are compressed at a force rate of $10^5$ μN/s. Figure 6a–c shows the atomic configuration of a cross-section of the 1:7 sample at 7%, 23%, and 53% strain. At 7% strain, it can be seen that failure and subsequent plasticity occurred via nucleation of partial

dislocations from the outer and inner surfaces of the nanobox and extended stacking faults (red atoms) that traverse the sidewalls of the nanobox. Several intersections have formed between dislocations that are nucleated at opposing walls. At 23% strain, dislocation nucleation continues at the top and bottom walls of the nanobox due to a bending moment from the deformation of the sidewalls. Dislocation density in the sidewalls increases, mainly in the central region. At 53% strain, defects left behind from dislocation slip and interactions including stacking faults and disordered regions are found throughout the entirety of the sidewalls, and the deformed sidewalls have contacted the top and bottom of the nanobox, which leads to rapidly increasing stress. Ag nanoboxes with wall thickness to length ratios of 1:16, and 1:25 were also investigated, to determine deformation mechanisms in nanoboxes with more slender sidewalls than the experimental nanoboxes. Figure 6d, e shows the atomic configuration of a cross-section of the 1:16 and 1:25 sample at 23% strain, respectively. The nucleation and glide of partial dislocations contribute to deformation in these geometries as well. With decreasing wall thickness, a greater amount of bending is observed in the sidewalls, and fewer of the extended stacking faults intersect. The observed bending agrees with the post compression SEM image (Fig. 4a), but is not observed in in situ TEM compression (Fig. 4c, and Supplementary Movie 1). This indicates that the loading state for the in situ SEM test is closer to ideal compression as in MD simulations, while the in situ TEM test may deviate further from ideal conditions. However, a failure of the sidewall is observed during the in situ TEM video (Supplementary Movie 1), which matches with the failure observed in MD simulations (Fig. 6c).

Interfaces such as twin boundaries and grain boundaries can act as barriers to dislocation motion and contribute to strengthening[50]. Previous MD simulations subjected nanotwinned Au nanowires to tension and reported significant strain

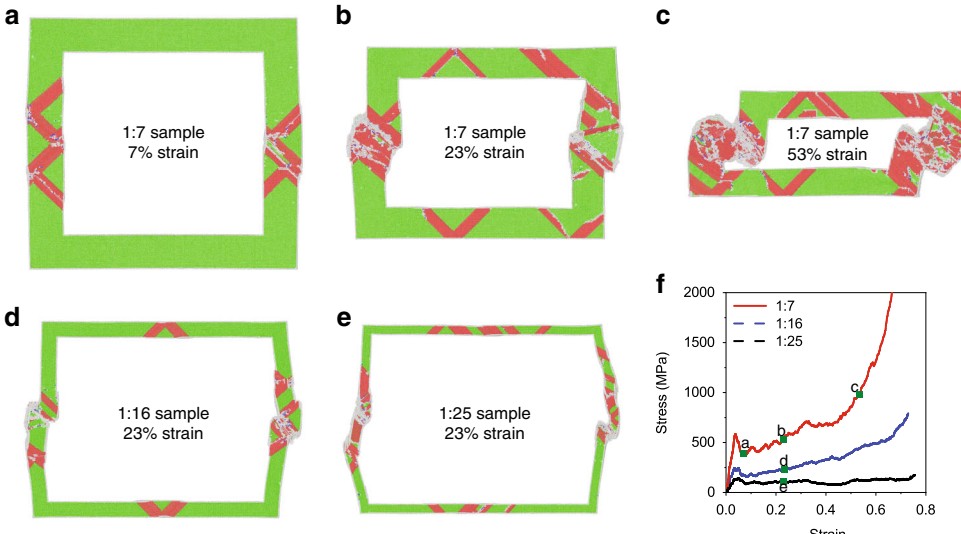

**Fig. 6 MD simulations of Ag nanobox.** Cross-sectional view of 1:7 sample (ratio of wall thickness to length) at **a** 7% strain, **b** 23% strain, and **c** 53% strain. Cross-sectional view of **d** 1:16 sample and **e** 1:25 sample at 23% strain. Green atoms are in fcc positions, red atoms are at stacking faults, and gray atoms are non-lattice atoms (either surface atoms or atoms with highly deformed local environment). **f** Compressive stress-strain curves for simulated samples of different geometries. The locations that correspond to the MD snapshots in a–e are marked in the stress-strain curves.

hardening that was attributed to coherent twin boundaries serving as strong barriers to the glide of leading partial dislocations[51]. This phenomenon has also been observed in MD simulations of nanotwinned Cu samples where twin boundaries were observed to have a strong blocking effect on dislocations[52]. We note that both of these studies employed interatomic potentials with similar low stacking fault energies, $43.4 \, mJ/m^2$ for Au potential[51] and $44.42 \, mJ/m^2$ Cu potential[52] which compares to the stacking fault energies of $6.24 \, mJ/m^2$ [53] and $26 \, mJ/m^2$ [54] of potentials used in this study. The energy barrier for the transmission of a partial dislocation across a twin boundary has been calculated to be ~4 meV/atom for FCC Au, and independent of twin size[55]. The energy barriers for dislocation slip transmission across a twin boundary is reported to be between 199 and $232 \, mJ/m^2$, which is similar to the lower bound of the energy barrier for grain boundary transmission[56]. Although similar values are not available for FCC Ag, the similarities between Au, Cu, and Ag indicate that the twin boundaries can contribute to strengthening. However, no full dislocations were observed in any MD compression sample. Despite the presence of extensive partial dislocations, formation of twins was rarely observed. Incipient twins of 3–4 atomic layers were observed, but deformation twinning was not a dominant deformation mode (Supplementary Fig. 1). In addition, since we stipulate that the nanoboxes are single crystalline, grain boundary hardening is not an available deformation mechanism. Deformation dominated by dislocation nucleation has also been reported in MD simulations of Cu ellipsoidal nanoshells[57] as well as partial dislocation and stacking fault dominated deformation in Pd nanoshells[58].

Deformation of the simulated smooth nanobox samples led to the stress strain-curves shown in Fig. 6f. In each of the samples, initial elastic deformation was followed by a local peak in stress, and then a stress drop that corresponds to the start of plasticity. All samples then undergo further plastic deformation followed by a rapid increase in stress due to contact between walls of the nanobox. In the stress-strain curve for the 1:7 sample, significant hardening is observed. This is due to extended stacking faults that act as barriers to dislocation motion (Fig. 6a-c). This can form a trapped leading partial dislocation that exerts a backstress that prevents the nucleation of its trailing partial and other

dislocations from the same surface source. This effectively shuts down active nucleation sources and requires nucleation from new sources for continued plasticity. Such a hardening mechanism is similar to that observed in both experiments and simulations of nanoporous Au[17,59–61]. Less hardening is observed in the 1:16 sample, and no hardening is observed in the 1:25 sample. There are fewer dislocations in these samples with more slender sidewalls, which provides further evidence that hardening is due to the proposed mechanism in which stacking faults impede dislocation glide.

Strain-hardening exponents were not computed for the generated MD stress-strain data. The high strain-rate of the simulated samples results in a much larger yield stress (and consequently overall stress level) than experiments, such that the simulated and experimental hardening rates cannot be quantitatively compared.

The effect of surface roughness on nanobox deformation and failure was also studied using MD simulation. Hemispherical protrusions of 5 nm diameter were added to the smooth nanobox to construct the rough nanobox (see Supplementary Fig. 2). The most notable difference between the stress-strain curves of the rough and smooth samples is a significant decrease in yield strength in the rough sample, suggesting a stress concentration factor of ~2. Both samples yield via partial dislocation nucleation from the surface. This suggests that the protrusions act to promote nucleation of dislocations, likely as stress concentrators. After an initial regime of negligible hardening, the simulated rough samples display hardening rates similar to the simulated smooth samples, which agrees with experiment. We note that the particular choice of protrusion distribution does not result in contact between protrusions during compression. The MD results are reasonable when compared with the expected stress concentrations from geometric considerations. Stress concentration estimated using classical mechanics equations for cylindrical shafts on a surface with fillet radius of curvatures ranging between 1–3 nm at the base and shaft diameters ~10 nm give a distribution of stress concentration factors of ~1.58[62]. Experimental yield strengths of rough and smooth nanoboxes indicate a stress concentration factor of ~1.35, which is within the range of the estimate. The presences of bumps and notches can increase

surface areas and alter the corresponding surface free energy. However, previous MD simulations performed on tensile testing of Silicon nanowires with notched rough surface also show that surface roughness stress concentrations dominate the yield strength over altered surface free energy[63]. In addition, no strain hardening is observed in these tensile loaded wires.

In conclusion, Au–Ag nanoboxes were compressed using in situ SEM and TEM uniaxial compression testing. Smooth nanoboxes were found to yield at $130 \pm 45$ MPa, and rough nanoboxes were found to yield at ~$96 \pm 31$ MPa. Both types of nanoboxes show significant hardening after yield, but the rough nanoboxes have a higher hardening rate. FEM simulations showed that the nanobox structural geometry does not lead to hardening. Based on MD simulations, we attribute the appearance of strain hardening in our experimental results to the structures' material response involving accumulation of crystalline defects such as dislocations and stacking faults within the walls of the nanoboxes. These results demonstrate the ability to simultaneously control yield strength and hardening by changing sample dimensions, while maintaining the same structural geometry. The mechanical properties of the nanoboxes indicate that they could be used as components of larger, hierarchical materials, which could be achieved through colloidal self-assembly. It would also be of interest to study nanoboxes with a range of wall thicknesses, in order to determine transitions in deformation mechanisms at extremely small sizes where dislocations no longer control mechanical properties.

## Methods

**Nanobox synthesis and structural characterization**. AgNO$_3$ (>99.9%, Fisher Scientific), Ethylene glycol (EG, Cl <5 ppm, and Fe <0.2 ppm, Macron Fine Chemicals), polyvinylpyrrolidone (PVP, Molecular weight of 55 000 g/mol, Sigma Aldrich), and HCl (37%, Fisher Scientific), are used in the polyol synthesis of solid Ag nanocubes. HAuCl$_4$ (>99.9%, Fisher Scientific) is used in the galvanic replacement reaction to form hollow nanoboxes.

Ag nanocubes are synthesized in EG. PVP is used as a surfactant. Solutions of 200 mM AgNO$_3$ in EG and 120 mM PVP in EG are prepared by sonication. A total of 60 mM HCl in EG is prepared by dissolving 4.9 μl of 37% HCl in 1 ml of EG. 1 ml of EG is heated in a 22 ml vial submerged in a mineral oil bath at 137 °C for 20 min partially covering the mouth with the cap. A total of 50 μl of HCl, 1 ml of the prepared AgNO$_3$ and 1 ml PVP solutions are added and the cap is closed when the reaction solution turns pink. The reaction is left at 137 °C for 2.5 h before the seeding starts, after which the reaction completes within 15–20 min resulting in a yellowish-green solution color. A total of 500 ml of the prepared Ag nanocubes are cleaned using centrifugation once with 2X amount of acetone to remove EG and three times with water to remove excess surfactant at 2500 rpm for 7 min and resuspended in 100 μl of water for further use in galvanic replacement.

A total of 100 μl of the cleaned Ag nanocubes are slowly reacted with 1.5 ml of 1 mM HAuCl$_4$ added drop wise over 30 min to obtain smooth Au–Ag nanoboxes with corner pits of 19 ± 4 nm diameter. Another batch of 100 μl of Ag nanocubes is reacted with 2.5 ml of HAuCl$_4$ added dropwise in 25 min to obtain the roughened Au nanoboxes. Both the samples are cleaned three times with water at 2500 rpm for 5 min to remove excess surfactant. Cleaned samples are deposited on silicon wafer for SEM imaging and in situ compressions as well as TEM grid for characterization (Fig. 2). SEM and bright field TEM imaging are used to characterize the samples for their lengths, wall thickness, and surface roughness.

TEM images were obtained on an FEI Tecnai G2 F20 X-TWIN TEM at 200 kV. FEI Helios NanoLab 600i DualBeam FIB/SEM is used to obtain SEM images for pre compression and post compression characterization. ImageJ is used to calculate particle sizes, wall thickness, roughness and hole sizes, etc. from the SEM and TEM images. The values reported are the average of measured sizes with the standard deviation of the set as +/– bounds.

**Mechanical testing**. In situ SEM mechanical testing was performed with the Nanoflip in situ mechanical tester (Nanomechanics, Inc.) inside of a FEI Helios NanoLab 600i DualBeam FIB/SEM. The nanoboxes were compressed at a constant loading rate of 1 μN/s with a 1000 Hz data acquisition rate. Nanoboxes were compressed up to ~60% strain. Force displacement curves were corrected for frame stiffness. Engineering stress and strain were computed by dividing the corrected force and displacement by the nominal (footprint) area and length of the structure, respectively. The strain hardening exponents were obtained by fitting an exponential relation of $\sigma = K\varepsilon^n$ to post-yield experimental data for each individual test curve. The values of $n$ from all tests were then averaged to get the reported values. The densification stage was not taken in the curve fitting as this sharp rise in stress

is due to touching sidewalls and not the material strain hardening effect. The fitting was limited up to 50% strain. The values of yield stress and strain hardening exponent reported are the average of measured sizes with the standard deviation of the set as +/– bounds.

In situ TEM compression testing was performed at the Center for Integrated Nanotechnologies, Sandia National Laboratories using a PI-95 nanoindenter (Bruker Inc.) in a JEOL 2100 TEM at 200 kV. Hollow Ag nanocubes were dropcast onto a 1 μm plateau Si wedge and compressed at a strain rate of 0.03 1/s.

**Finite element modeling**. A 3D model of the hollow nanobox was created in SolidWorks, and then exported to Abaqus/CAE 2018. The nanobox was seeded at the edges with a global spacing of 7 and then meshed with tetrahedral elements, resulting in over 400,000 nodes and 200,000 elements. An isotropic and anisotropic elastic perfectly plastic material model was used. For the isotropic model, an elastic modulus and yield strength of 47 GPa and 600 MPa were used, respectively, which were derived from Ag <100> properties. For the anisotropic model, the elastic stiffness matrix was populated using $c_{11}$, $c_{12}$, and $c_{44}$ constants for silver of 124, 93.4, and 46.1 GPa, respectively (See Supplementary Methods – FEM simulation)[64]. The nanobox was placed between a rigid bottom plate that was fixed in all degrees of freedom and a rigid top plate that was only allowed to move in the z-direction (loading direction). Load was applied normal to the top plate to simulate compression. The static Riks (or arc length) method was used to account for the possibility of negative or near-zero stiffness during compression.

**Molecular dynamics simulations**. MD simulation samples were constructed by starting with a solid cube of Ag atoms with dimensions of 50 nm and {100} faces. Interior atoms were removed to achieve desired wall thickness to length aspect ratios. For all geometries, a 7 nm wide cube was removed from each of the corners of the simulated hollow nanobox (Supplementary Fig. 3). Before compression, all samples were subjected to energy minimization and thermal relaxation. The total potential energy of the initial sample configuration was minimized following a conjugate gradient scheme allowing the atomic positions to relax. The system was then assigned a velocity profile corresponding to 300 K and the samples were thermally equilibrated under time integration of the NVT ensemble for 200 ps. The samples were then equilibrated for an additional 200 ps using Langevin dynamics in order to dampen the pressure oscillations caused by rapid application of temperature. Compression was applied to the simulated nanoboxes using a force-controlled scheme by creating two stiff Si plates above and below the nanobox. The Ag-Si interaction used to simulate the interaction between the compression plates and the nanobox was assumed to be of a Lennard-Jones 6–12 type with interaction parameters from Ngandjong et al.[65]. During compression, the bottom plate is held fixed while the top plate is subjected to a constant force rate. In order to investigate the rate effect in simulated compression, we subjected all nanobox samples to force rates of $10^5$, $10^6$, $10^7$, and $10^8$ μN/s (Supplementary Fig. 4). Minimal differences in both the output atomic configurations and the stress-strain data between the $10^6$ μN/s and $10^5$ μN/s simulations indicate that rate effects are minimized in the $10^5$ μN/s. We, therefore, report the results of the lowest force rate $10^5$ μN/s in the manuscript. Simulated compression was performed until >80% strain unless non-physical behavior was observed in the atomic output, in which case the simulation was halted. Interactions between Ag atoms were modelled using the embedded atom method (EAM) and implemented using a potential developed by Foiles et al.[53]. In order to check the validity of this Ag potential, we repeated compression simulations using an Ag potential from Sheng, et al. which was fit to a different set of physical properties including the stable and unstable stacking fault energies[54]. The Foiles potential resulted in values of 6.24 mJ/m$^2$ and 116 mJ/m$^2$, and the Sheng potential resulted in values of 26 mJ/m$^2$ and 114 mJ/m$^2$ for stable and unstable stacking fault energies, respectively. Quantitatively, the stress-strain curves generated using both potentials and the resulting atomic deformation mechanisms are similar (Supplementary Fig. 5). This shows that the Foiles potential can capture the relevant deformation behavior in the Ag nanoboxes, which are the simulations that are shown in the manuscript. All simulations were performed using the LAMMPS molecular dynamics software[66] using a timestep of 1 fs and OVITO[67] was used for common neighbor analysis (CNA) and visualization of atomic configurations.

To calculate the stress in the simulated nanobox, we relate engineering stress to the atomic quantities that are computed during MD simulations. For compression aligned along the z-axis, stress is determined as shown in below equation:

$$\sigma_{zz} = \frac{F_z}{A_0} = \frac{\sum vs_{zz}/L_z}{L_{x0}L_{y0}} \tag{3}$$

where $L_{x0}$ and $L_{y0}$ are the initial cross-sectional dimensions of the nanobox, $L_z$ is the height of the nanobox, $v$ is the atomic volume, $s_{zz}$ is the atomic stress and the summation is over all nanobox atoms. The product, $vs_{zz}$, can be computed for each nanobox atom following standard definitions[68]. $L_z$ is taken as the separation distance between the two rigid plates and $L_{x0}$ and $L_{y0}$ are computed at the beginning of the simulation using the extrema atomic positions. This form was chosen to most closely match the computation of engineering stress in experiments. The numerator in Eq. (3) uses the standard calculation of virial stress using the instantaneous sample configuration and is analogous to the instantaneous force as measured during compression experiments. In order to match the

experimental calculation of engineering stress, we then divide by the initial cross-sectional area of the nanobox.

## Data availability
The data that support the findings of this study are available from the corresponding author upon reasonable request.

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

## Acknowledgements

We gratefully acknowledge financial support from the Stanford SystemX Alliance. M.T.K. is supported by the National Defense and Science Engineering Graduate Fellowship. D.D. is supported by the National Science Foundation Graduate Research Fellowship under Grant No. 1656518. C.M.B. and K.H. time was fully supported by the Division of Materials Science and Engineering, Office of Basic Energy Sciences, U.S. Department of Energy. Part of this work was performed at the Stanford Nano Shared Facilities (SNSF), which is supported by the National Science Foundation under award ECCS-1542152. This work was performed, in part, at the Center for Integrated Nanotechnologies, an Office of Science User Facility operated for the U.S. Department of Energy (DOE) Office of Science. Sandia National Laboratories is a multimission laboratory managed and operated by National Technology & Engineering Solutions of Sandia, LLC, a wholly owned subsidiary of Honeywell International, Inc., for the U.S. DOE's National Nuclear Security Administration under contract DE-NA-0003525. The views expressed in the article do not necessarily represent the views of the U.S. DOE or the United States Government. We thank Prof. Ill Ryu for insightful discussion regarding underlying mechanisms in rough nanobox deformations. We gratefully acknowledge the financial support from the Agency for Science, Technology and Research (A*STAR) under grant AMDM A1898b0043, and the use of computing resources at the A*STAR Computational Resource Centre, Singapore.

## Author contributions

R.P.P. and X.W.G. designed the study. R.P.P. synthesized the nanostructures, R.P.P. and M.T.K. and conducted in situ SEM mechanical tests, M.T.K. imaged the nanostructures for characterization, M.T.K. conducted in situ TEM tests with C.M.B. and K.H., R.P.P. and X.W.G. analyzed the experimental data, D.D. performed the finite element modelling, Z.H.A., S.C. and Y.W.Z. performed the molecular dynamics simulations. R.P.P. wrote the manuscript under supervision of X.W.G. with contribution from co-authors for corresponding sections.

## Competing interests

The authors declare no competing interests.
