## [Peer Review File · Nature Communications]

Reviewers' Comments:

Reviewer #1:

Remarks to the Author:

This paper discusses the mechanical properties of metallic nanoboxes including both yield stresses and strain hardening behaviors. The topic is interesting, and the material system is new. The authors took multiple steps (compression test, FEM, MD) to investigate the mechanisms. However, the study sounds incomplete and many suggested mechanisms were based on authors' assumption and not rationalized in detail by any of the simulation or experimental methods. Therefore, the reviewer believes that this paper needs substantial revision and cannot be recommended for publication. Following are some questions that should be addressed in better details.

- The nanobox with rough surfaces has much lower yield point. The reason behind this is not discussed in detail. The authors could possibly look into dislocation nucleation mechanisms of both rough and smooth surfaces and identify the sources of this difference. A detailed discussion should be devoted to the existences of the steps and ledges in the rough surface and how their distribution and orientation could affect the dislocation nucleation to be easier in the case of rough surfaces.

- In addition, the size of the rough and smooth boxes is different. Smooth nanoboxes are about 10% larger than the rough ones. The yield differences could simply be related to the size effects. The authors are recommended to address this phenomenon.

- Discussions on volume differences in page 6 are unclear and puzzling. The language of this discussion needs to be improved to address the main point of all these volume comparisons.

- Greater rate of hardening could be related to larger surface ratios. As the scale of roughness is similar to structure dimensions. This should also be discussed from both TEM and MD aspects. The authors only speculated two possible scenarios without backing up their hypothesis.

- Page 8 the following statement can be discussed in more details. In what terms are the similarities and differences. "The mechanical response of the hollow nanoboxes is substantially different from that of solid, single crystalline metallic nanostructures, but is similar to nano- and microporous metal foams, and thin-shelled nanostructures such as carbon nanotubes".

- Page 9. It is not clear if the deformation mode of the sidewalls includes any bending and buckling or they are just under compression. The authors first suggested buckling as a possible deformation mode. MD simulations also show signs of bending and buckling. However, the experimental observations decline any evidence of bending and buckling. This discrepancy and the major deformation modes need to be discussed in better details.

- In the methods section why the potential for Ag-Si is needed? There is no Si atoms in the system!

Reviewer #2:

Remarks to the Author:

Authors study hollow nanocubes made of Ag and Au, comparing the mechanical behavior under compression for nanocubes with smooth and rough surfaces. Hollow cube synthesis required several steps, resulting in cube sides about 140 nm with wall thickness around 15 nm. Some of the authors carried out a previous study, their ref. [42], compressing Ag nanocubes which are passivated with surfactants to tune surface properties which would affect dislocation nucleation. This new study also includes high resolution TEM with in situ nanomechanical testing, finite

element methods (FEM), and molecular dynamics (MD) simulations. It is concluded that the strong hardening at large compression, which is observed experimentally, is due to stacking fault-dislocation interactions.

The behavior of nanostructures depends on size, as discussed for instance in [Li 2018, Wang 2018], showing diffusion-dominated plasticity dominating below around 10 nm [Sun 2014], and dislocation slip [Shong 2016, Wang 2018a] and twinning [Wang 2018b] above that size. For nanocrystalline fcc metals a similar set of regimes appears with increasing grain size, with GB activity replacing surface diffusion mediated plasticity [Li 2018b]. There is a massive amount of research on nanowires but research on mechanical properties of metallic hollow structures is scarce, despite the possibility of technological applications. This paper is a welcomed contribution to the field of nanomechanics, but some of the conclusions are not convincing for this reviewer. A summary of results is provided below, along some of the caveats which can weaken the conclusions.

Regarding the experiment, loading curves under the TEM give a yield stress of approximately 130 and 95 GPa and 10% strain, for smooth and rough nanocubes, and deformation which can be as large as 60%-80%. TEM suggest the nucleation of dislocations during compression, but scale bar is 100 nm in Fig. 4, so individual dislocations cannot be resolved, as they are in some other studies for Ag nanostructures [Sun2014]. There is significant hardening starting at around 40% for the smooth cube and 20% for the rough cube. The comparison between smooth and rough is complicated by the fact that the density, chemical composition, and small-scale topology of the cubes are different.

Regarding FEM simulations, an isotropic elastic-plastic solid is assumed, with a bulk yield strength of 600 MPa, which gives nanocube yielding at around 230 MPa and 2% strain, and a stress plateau (with slight softening) after that. The authors indicate that, since FEM does not show hardening, materials rather than geometry determine experimental hardening. However, in order to obtain hardening, one has to include certain ingredients in FEM calculations, and it is not clear if that is the case. Was any type of hardening exponent used for the plastic behavior? In addition to this, FEM has to be applied out with care at the nanoscale: anisotropic crystal plasticity and plasticity nucleation effects [Li 2002] might be required. Typical representative volume elements might be larger than the nanostructure representative size (for instance cube side here). It would be useful to demonstrate that the FEM simulations used do have the ingredients required to produce hardening for some given wall thickness.

Finally, MD simulations of the hollow nanocube are scaled down with respect to the experiment, using a cube side of 50 nm, and 3 different values of thickness: 7, 3, and 2 nm. They assume Ag as the single component of the cube. MD simulations are carried out at different compression rates, to test for strain rate effects, given that experiments are carried out at a much longer time scale. They give, for the sample with 1:7 ratio of wall thickness to cube length, a plastic yielding near 585 MPa and 4% strain, with strong hardening associated with densification when walls start to touch due to compression after 48% strain. Snapshots of the simulations indicate stacking faults, but there is no detailed dislocation analysis. For instance, the paper says "At 53% strain, dislocations are found throughout the entirety of the sidewalls". Fig. 6c shows evidence for stacking faults and some disordered regions, but most partial dislocations would have disappeared leaving behind only stacking faults. Full dislocations are unlikely to appear for such thin wall. Is there any twinning in the MD samples?

Regarding simulation of hollow nanostructures of fcc metals, there are two recent studies: one focusing on elastic properties of Cu nanoshells [Yang 2018], and another including a study of plastic properties for Pd nanoshells [Valencia2018]. Pd has much higher SFE, but still shows mostly SFs and twins at high strain. A possible problem with the MD simulations presented in the paper is the use of an interatomic potential not suited for plasticity under compression up to large strains. Mordehai et al [Mordehai2018] showed that the plastic deformation of a solid nanocube

depends strongly on the potential used. There are many potentials for Ag [OpenKim2019]. The EAM potential used in this work [Foiles 1986] was not fit to describe properties related to dislocation plasticity, unlike more recent potentials like the one by Sheng [Sheng 2011], used in [Sun 2014]. A recent study [Rassoulinejad-Mousavi 2018] shows that elastic constants under pressure are not well described by this Ag potential. Another recent study [Su 2019] shows that the [Sheng 2011] potential is the one which compares better with ab-initio calculations of the generalized stacking fault energy surface, which is one of the main factors in stacking fault nucleation and motion. The potential by Foiles [Foiles 1986] gives a much lower value for the unstable and stable stacking faults. Therefore, the observed strong hardening might be partly due to the (artificially) large number of stacking faults due to this lower barrier.

There are always challenges when comparing experiments with different numerical techniques. In this case, the comparison does not provide quantitative agreement and improved simulations and analysis would be welcomed.

References:

- [Foiles 1986] S.M. Foiles et al., Embedded-atom method functions for the fcc metals Cu, Ag, Au, Ni, Pd, Pt, and their alloys, *Phys.Rev. B* 33, 7983 (1986).
- [Li 2002] J. Li et al., Atomistic mechanisms governing elastic limit and incipient plasticity in crystals, *Nature* 418, 307 (2002).
- [Li 2018] Qing-Jie Li & Evan Ma (2018) When 'smaller is stronger' no longer holds, *Materials Research Letters*, 6:5, 283-292.
- [Li 2018b] Duohui Li, Xinyu Shu, Deli Kong, Hao Zhou, Yanhui Chen. Revealing the atomistic deformation mechanisms of face-centered cubic nanocrystalline metals with atomic-scale mechanical microscopy: A review. *Journal of Materials Science & Technology* 2018, 34 (11) , 2027-2034. DOI: 10.1016/j.jmst.2018.03.006.
- [Mordehai 2018] D. Mordehai et al., Nucleation-Controlled Plasticity of Metallic Nanowires and Nanoparticles, *Adv.Mater.* 2018, 1706710.
- [OpenKim2019] <https://openkim.org/browse/models/by-species?species-search=Ag>
- [Rassoulinejad-Mousavi 2018] S.M. Rassoulinejad-Mousavi & Y. Zhang, Interatomic Potentials Transferability for Molecular Simulations: A Comparative Study for Platinum, Gold and Silver, *Scientific Reports* 8, 2424 (2018).
- [Sheng 2011] H.W. Sheng et al., Highly optimized embedded-atom-method potentials for fourteen fcc metals, *Phys. Rev. B* 83, 134118 (2011). <https://sites.google.com/site/eampotentials/Ag>
- [Shong 2016] L. Shong et al., Slip-activated surface creep with room-temperature super-elongation in metallic nanocrystals, *Nature Mat.* 16, 439 (2016).
- [Su 2019] Y. Su et al., Density functional theory calculations of generalized stacking fault energy surfaces for eight face-centered cubic transition metals, *J. Appl. Phys.* 126, 105112 (2019); <https://doi.org/10.1063/1.5115282>
- [Sun 2014] Sun et al. 2014, Liquid-like pseudoelasticity of sub-10-nm crystalline silver particles, *Nature Materials* 13, pages 1007–1012 (2014).
- [Valencia 2018] F. Valencia et al., Mechanical Properties Obtained by Indentation of Hollow Pd Nanoparticles, *J. Phys. Chem. C* 122, 25035-25042 (2018).
- [Wang 2018] X Wang et al., "Advances in understanding atomic-scale deformation of small-sized face-centered cubic metals with in situ transmission electron microscopy", *Materials Today Nano* 2, 58 (2018).
- [Wang 2018a] L Wang et al., Plastic Deformation through Dislocation Saturation in Ultrasmall Pt Nanocrystals and Its in Situ Atomistic Mechanisms, *Nano Lett.* 20171784733-4739
- [Wang 2018b] L. Wang, J. Teng, Y. Wu, J. Zou, G. Yu, Z. Zhang, X. Han. Size dependence of dislocation activities and independence on theoretical elastic strain limit in Pt nanocrystals revealed by atomic-resolution in situ investigation. *Materials Today Nano* 2018, 2, 1-6. DOI: 10.1016/j.mtnano.2018.04.007.
- [Yang 2018] Q. Yang et al., Mechanical properties of single crystal copper ellipsoidal nanoshells by molecular dynamics, *International Journal of Modern Physics B* Vol. 32, No. 16, 1850196 (2018). <https://doi.org/10.1142/S0217979218501965>.

Eduardo Bringa (ebringa@yahoo.com)

Reviewer #3:

Remarks to the Author:

This manuscript presents a combined experimental and computational study of plastic deformation and strain hardening during compression of hollow Au-Ag nanoboxes fabricated by galvanic replacement from perfect solid Ag nanocubes. The authors have tested metallic nanostructures of ~ 100 nm in size made with either smooth or rough surfaces. Using in situ SEM/TEM compression experiments, they observe that these nanoboxes exhibit strain-hardening behaviors exceeding that predicted theoretically for bulk Ag, more so when the surface of nanoboxes is rough as opposed to smooth. FEA and MD simulations were used to study the mechanism responsible for this behavior. It is concluded that the observed hardening effects result from dislocation interaction with stacking-fault (SF) defects, rather than from structural plastic buckling effects. This research is new and interesting, and the article is well-written and well-presented. However, the main claim that the governing mechanism is associated with dislocation-SF interactions should be better substantiated before this manuscript could be accepted for publication in Nature Communications. In particular, the authors should address the following major concerns:

1. The authors did not clearly address the internal microstructure differences between smooth and rough nanoboxes, such as the presence of grain boundaries, which raises an important concern that grain-boundary hardening should not have been omitted in the models.
2. The author should better address how dislocation-SF interactions could significantly contribute to the overall hardening and how they are truly impacted by roughness effects. Particularly, experimental and MD simulation evidence suggests that dislocation interactions with pre-existing SFs are associated with very small transmission energy barriers (e.g. see Wang et al. *Nanoscale*, 2019, 11, 12672). Therefore, the authors should propose a better model to explain the observation that rough nanoboxes exhibit superior hardening. In its current version, the manuscript remains unsettled on this question.
3. FEA simulation results presented are not meaningful and should be further developed because (1) they focused on only plastic buckling effects at strains $< 20\%$, while structural effects in MD simulations become only evident at strains $> 40\%$, and (2) the elastic-plastic constitutive law used for FEA is unclear. In fact, the authors should test their hypothesis by simulating constitutive laws with different hardening exponents to study how it could affect the strain-strain response of metallic nanoboxes. These results would be more insightful.
4. The authors must clarify the following technical aspects in the text: (1) how stress was computed in each method, especially in MD simulations, (2) how strain-hardening exponents were measured experimentally, and if the last densification stage where stress increases sharply was taken into account or not, and (3) if strain-hardening exponent was quantified in MD simulations, and that such prediction should be compared to the experimental exponents.

Response to Reviewer Comments for

Hardening in Au-Ag Nanoboxes from Stacking Fault-Dislocation Interactions

Radhika P. Patil^a, David Doan^a, Zachary H. Aitken^c, Shuai Chen^c, Mehrdad T. Kiani^b,
Christopher M. Barr^d, Khalid Hattar^d, Yong-Wei Zhang^c, X. Wendy Gu^{*a}

^aDepartment of Mechanical Engineering, and ^bDepartment of Materials Science and Engineering,
Stanford University, Stanford, CA 94305, United States

^cInstitute of High Performance Computing, A*STAR, 1 Fusionopolis Way, #16-16 Connexis,
138632 Singapore

^dMaterials, Physical, and Chemical Sciences, Sandia National Laboratories, Albuquerque, NM,
87185, USA

Reviewer #1 comments:

This paper discusses the mechanical properties of metallic nanoboxes including both yield stresses and strain hardening behaviors. The topic is interesting, and the material system is new. The authors took multiple steps (compression test, FEM, MD) to investigate the mechanisms. However, the study sounds incomplete and many suggested mechanisms were based on authors' assumption and not rationalized in detail by any of the simulation or experimental methods. Therefore, the reviewer believes that this paper needs substantial revision and cannot be recommended for publication. Following are some questions that should be addressed in better details.

- *The nonbox with rough surfaces has much lower yield point. The reason behind this is not*
*discussed in detail. The authors could possibly look into dislocation nucleation mechanisms of*
*both rough and smooth surfaces and identify the sources of this difference. A detailed discussion*
*should be devoted to the existences of the steps and ledges in the rough surface and how their*
*distribution and orientation could affect the dislocation nucleation to be easier in the case of*
*rough surfaces.*

Response: We hypothesize that the rough nanoboxes yield at a lower stress because the rough
protrusions behave as stress concentrations. We agree that more careful characterization of the
surface roughness on the nanoboxes would be desirable for this analysis, but unfortunately it is
nearly impossible to do this using TEM, which is the highest resolution imaging technique that is
available to us. Due to the high density of protrusions, a TEM image oriented along the [100]
face of the nanobox shows the superposition of many different protrusions. This means that it is
impossible to estimate the shape of a single protrusion from this TEM image. To address this
issue, we were able to synthesize nanocubes with sparse protrusions by lowering the etchant
concentration. Figure 1 (below) shows TEM images of nanocube samples with low-density
protrusions, which were used to estimate the size of individual protrusions. Figure 1b shows a
high resolution TEM image of the corner of a nanocube, which shows lattice fringes within the
protrusion that are aligned with the crystal orientation of the cube. This indicates single
crystallinity and the lack of grain boundaries at the protrusion.

Figure 1. **a** TEM image of solid Ag nanocubes with sparse surface roughness. **b** HRTEM image
 of individual surface protrusion.

Based on these images, we estimate that protrusions with a radius of curvature (fillet radius) of
 $\sim 1\text{-}3$ nm at the base and diameters of ~ 10 nm. Using the classical stress concentration
 corresponding to a shoulder fillet in stepped circular shafts (Fig. 2) and the dimensions of the
 protrusions resulted in stress concentrations of ~ 1.58 .¹ Experimentally, the relative yield
 strengths of the rough and smooth nanoboxes indicate a stress concentration factor of ~ 1.35 for
 the rough nanoboxes, which is in the range of our estimate. This discussion has been added to the
 manuscript on page 14, line 286-290.

Fig. 2 Illustration of shoulder fillet in stepped circular shafts

We have also performed additional MD simulations of rough nanoboxes (nanoboxes with
 hemispherical protrusions) (see figure below). The most notable difference between the stress-
 strain curves of the rough and smooth samples is the significant decrease in yield strength in the
 rough sample. Both samples yield via partial dislocation emission from the surface. This
 suggests that the protrusions act to promote nucleation of dislocations, likely as stress
 concentrators, with a stress concentration factor of ~ 2 . We have included this discussion in the
 manuscript on page 13, line 275-286 and the following figure in the Supplemental Information.

 Fig. 3. MD simulations of Ag nanobox with rough protrusions and indents. **a** The rough nanobox
 sample. **b** Comparison between the generated stress-strain data of smooth and rough nanoboxes.
 Cross-sectional view of 1:7 rough sample (ratio of wall thickness to length) at **c** 4% strain, **d** 7%
 strain, **e** 25% strain, and **f** 40% strain. Green atoms have FCC coordination and red atoms are
 stacking faults.

- *In addition, the size of the rough and smooth boxes is different. Smooth nanoboxes are about*
*10% larger than the rough ones. The yield differences could simply be related to the size effects.*
*The authors are recommended to address this phenomenon.*

Response: While the overall size of the smooth nanoboxes is larger than the rough nanoboxes,
the characteristic length scale of interest is the thickness of the sidewall, as this is the load-
bearing structure within the nanobox. The rough nanoboxes have thinner walls than the smooth
nanoboxes. Considering the “smaller is stronger” size effect in nanoscale single crystalline
metals², the thinner walls should be stronger than the thicker ones, which is not observed. Other
researchers have proposed a size-independent strength when metallic nanostructures are in the
200 nm regime.³ This would result in rough nanoboxes with the same strength as smooth
nanoboxes per nanobox cross-sectional (load-bearing) area. This is also not observed. In
addition, the overall size of the rough nanoboxes is smaller than the smooth nanoboxes. The
rough nanoboxes would be stronger than the smooth if strength scaled as overall nanobox size
according to the trend of “smaller is stronger”, but this is not the case. Therefore, we conclude
that differences in yield strength are not due to size effects. The main difference between the
rough and smooth samples is surface roughness, so we propose that this difference is responsible
for differences in yield strength and hardening.

This discussion has been added to the manuscript on page 8, line 149-161.

- *Discussions on volume differences in page 6 are unclear and puzzling. The language of this*
*discussion needs to be improved to address the main point of all these volume comparisons.*

Response: We apologize for the confusion and lack of clarity. The main point of the discussion is

that both Au and Ag are present in the final nanoboxes, which indicates that the galvanic
exchange reaction does not go to completion. The evidence for this is the differences between the
volume of the solid Ag nanocube (starting structure), smooth nanobox and rough nanobox. The
section has been modified to better convey this main point (page 6, line 111-122).

- *Greater rate of hardening could be related to larger surface ratios. As the scale of roughness is*
*similar to structure dimensions. This should also be discussed from both TEM and MD aspects.*
*The authors only speculated two possible scenarios without backing up their hypothesis.*

Response: We have performed additional MD simulations to address this issue (see figure
below). The most notable difference between the stress-strain curves of the rough and smooth
samples is the significant decrease in yield strength in the rough sample. Both samples yield via
partial dislocation emission from the surface. This suggests that the protrusions act to promote
nucleation of dislocations, likely as stress concentrators. Beyond yielding, there is a regime (~10-
20% strain) where the smooth sample displays significant hardening and the rough sample
displays negligible hardening. This difference appears to be related to the difference in location
of the dislocation nucleation sites. In the smooth sample, initial yielding involves nucleation of
multiple dislocations from the center of the side walls (manuscript Figure 6). These dislocations
easily encounter each other and interact. In the rough sample, initial yielding involves
dislocations that are spaced far away from each other and have no opportunity to interact. It is
only beginning at ~25% strain, when further dislocations are emitted and begin to interact does
the rate of stress increase in the rough sample begin to match that of the smooth sample. Further
straining indicates similar levels of hardening between the smooth and rough samples. Although
the hardening rate of the rough sample approaches and then matches the hardening rate of the

smooth sample, we note that the distribution of protrusions that we use in our MD sample do not
contact each other throughout the entire compression simulation. We have included this
discussion in the manuscript on page 14, line 283 and have included the following figure in the
Supplemental Information.

Fig. 4. MD simulations of Ag nanobox with rough protrusions and indents. **a** The rough nanobox
sample. **b** Comparison between the generated stress-strain data of smooth and rough nanoboxes.
Cross-sectional view of 1:7 rough sample (ratio of wall thickness to length) at **c** 4% strain, **d** 7%
strain, **e** 25% strain, and **f** 40% strain. Green atoms have FCC coordination and red atoms are
stacking faults.

We agree with the reviewer that the presences of surface protrusions will increase surface area
and alter the corresponding surface free energy. We have discussed the morphology of the

surface protrusions using TEM imaging in response to reviewer #1 comments (see Figure 1).
From this analysis, we can get an estimate of the ratio of surface protrusion base diameter (l) to
height (d). The average $l:d$ ratio of the roughness is 1. Previously, MD simulations have been
performed on tensile testing of silicon nanowires with notched rough surface for varying ($l:d$)
ratios.⁴ These simulations show that surface roughness-induced stress concentrations dominate
the yield stress. The effect of surface energy change due to presence of roughness was also
investigated in this work. The generated stress-strain plots of these wires show no post yield
hardening despite the roughness altered surface energies. On the contrary, stress drops after
yielding in these simulations. Therefore, we hypothesize that surface energy alterations are not a
dominant factor for the observed hardening in our nanoboxes. Rather it is the interaction of
stacking faults and dislocations due to bending of the side walls and multiple nucleation sites as
seen from the above MD simulations which was not possible in a tensile tested sample. This
discussion has been added to the manuscript on page 14 line 290-294.

- Page 8 the following statement can be discussed in more details. In what terms are the
similarities and differences. “The mechanical response of the hollow nanoboxes is substantially
different from that of solid, single crystalline metallic nanostructures, but is similar to nano- and
microporous metal foams, and thin-shelled nanostructures such as carbon nanotubes”.

Response: The compressive stress-strain responses of single crystalline metallic nanostructures
like nanocubes, nanowires and nanopillars are characterized by large slip events in the plastic
regime, and lack of hardening. Metallic foams and thin-shelled nanostructures have stress-strain
curves without large slip events, with hardening. This difference is primarily due to deformation
modes such as bending and buckling. This behavior (gradual deformation without large slip

events and hardening) is similar to that of the nanoboxes, while the behavior of the solid single
crystalline nanostructures (large slip events, no hardening) is different from that of the
nanoboxes. This discussion has been added to the manuscript on page 9, line 164-172.

- Page 9. It is not clear if the deformation mode of the sidewalls includes any bending and
buckling or they are just under compression. The authors first suggested buckling as a possible
deformation mode. MD simulations also show signs of bending and buckling. However, the
experimental observations decline any evidence of bending and buckling. This discrepancy and
the major deformation modes need to be discussed in better details.

Response: The nanoboxes are deforming in bending and plastic buckling. This is clearly shown
in MD simulation. Unfortunately, in-situ SEM compression videos cannot be obtained at high
enough resolution to observe bending/buckling as it occurs. But, post-compression SEM images
(Figure 4a) show a compressed, smooth nanobox that is wider than that expected from elastic
deformation alone. This indicates that bending or buckling must occur during compression to
result in this final shape. A discrepancy arises when MD simulations and the post-compression
SEM image are compared to the in-situ TEM movie of nanobox compression, which does not
show bending, but does show a gradual increase in nanobox width during compression. This
indicates that the loading state for the in-situ SEM test is closer to ideal compression as in MD
simulations, while the in-situ TEM test may deviate further from ideal conditions. However, we
see failure of the sidewall in the *in-situ* TEM movie that is very similar to failure in MD
simulation. In the manuscript, we explain this discrepancy as “Differences in the post-
compression *in-situ* SEM and TEM mechanical testing images may be due to differences in
strain rate, noise levels, or subtle differences in nanobox geometry. In addition, the experimental

stress-strain curves show a range of behaviors for both the smooth and rough nanoboxes, which
may result in different deformation modes.” We have clarified these issues in the manuscript by
revising the text on page 10, line 188 (section page 9,177-200) and rearranging and adding text
to page 12, line 236-241.

- *In the methods section why the potential for Ag-Si is needed? There is no Si atoms in the*
*system!*

Response: Our MD methodology involves compressing the nanobox between two stiff plates.

Due to availability of potentials, we chose to construct these plates out of Si atoms. These plates

interact with the Ag atoms via a Lennard-Jones potential, but do not interact with other Si atoms.

We have clarified the purpose of the Ag-Si potential in the methods section, page 18, line 382-

384.

**Reviewer #2 comments:**

*Authors study hollow nanocubes made of Ag and Au, comparing the mechanical behavior under*

*compression for nanocubes with smooth and rough surfaces. Hollow cube synthesis required*

*several steps, resulting in cube sides about 140 nm with wall thickness around 15 nm. Some of*

*the authors carried out a previous study, their ref. [42], compressing Ag nanocubes which are*

*passivated with surfactants to tune surface properties which would affect dislocation nucleation.*

*This new study also includes high resolution TEM with in situ nanomechanical testing, finite*

*element methods (FEM), and molecular dynamics (MD) simulations. It is concluded that the*

*strong hardening at large compression, which is observed experimentally, is due to stacking*

*fault-dislocation interactions.*

*The behavior of nanostructures depends on size, as discussed for instance in [Li 2018, Wang*
*2018], showing diffusion-dominated plasticity dominating below around 10 nm [Sun 2014], and*
*dislocation slip [Shong 2016, Wang 2018a] and twinning [Wang 2018b] above that size. For*
*nanocrystalline fcc metals a similar set of regimes appears with increasing grain size, with GB*
*activity replacing surface diffusion mediated plasticity [Li 2018b]. There is a massive amount of*
*research on nanowires but research on mechanical properties of metallic hollow structures is*
*scarce, despite the possibility of technological applications. This paper is a welcomed*
*contribution to the field of nanomechanics, but some of the conclusions are not convincing for*
*this reviewer. A summary of results is provided below, along some of the caveats which can*
*weaken the conclusions.*

*Regarding the experiment, loading curves under the TEM give a yield stress of approximately*
*130 and 95 GPa and 10% strain, for smooth and rough nanocubes, and deformation which can*
*be as large as 60%-80%. TEM suggest the nucleation of dislocations during compression, but*
*scale bar is 100 nm in Fig. 4, so individual dislocations cannot be resolved, as they are in some*
*other studies for Ag nanostructures [Sun2014]. There is significant hardening starting at around*
*40% for the smooth cube and 20% for the rough cube. The comparison between smooth and*
*rough is complicated by the fact that the density, chemical composition, and small-scale*
*topology of the cubes are different.*

*Regarding FEM simulations, an isotropic elastic-plastic solid is assumed, with a bulk yield*

*strength of 600 MPa, which gives nanocube yielding at around 230 MPa and 2% strain, and a*
*stress plateau (with slight softening) after that. The authors indicate that, since FEM does not*
*show hardening, materials rather than geometry determine experimental hardening. However, in*
*order to obtain hardening, one has to include certain ingredients in FEM calculations, and it is*
*not clear if that is the case. Was any type of hardening exponent used for the plastic behavior? In*
*addition to this, FEM has to be applied out with care at the nanoscale: anisotropic crystal*
*plasticity and plasticity nucleation effects [Li 2002] might be required. Typical representative*
*volume elements might be larger than the nanostructure representative size (for instance cube*
*side here). It would be useful to demonstrate that the FEM simulations used do have the*
*ingredients required to produce hardening for some*
*given wall thickness.*

Response: A perfectly elastic-plastic material model without strain hardening was used in the
FEM simulation. The reviewer is correct that more complicated material models are typically
required to represent nanostructures using FEM. The goal of the FEM simulations was actually
not to reproduce the experimental results (this is left to the MD simulations, which are more
appropriate for the size scale and complexity of the nanoboxes). Rather, the point of the FEM
simulations is to answer the question, “What is the structural response for a material model that
does not include strain hardening?” The reason that this is relevant to our system, is that the solid
Ag nanocube (the parent structure from which the nanobox is formed) does not show strain
hardening. It has been shown in previous work from our group that Ag nanocubes have a stress-
strain curve typical of single crystalline nanostructures, where there is no hardening after yield.⁵
Starting from this point, we postulate that because the hollow nanoboxes are simply the solid
nanocubes with the interior and corners removed, the same material model should be used for the

nanoboxes and solid nanocubes. For instance, the crystal anisotropy and surface dislocation
nucleation effects should remain the same for the hollow nanoboxes and solid nanocubes. Using
this approach results in a discrepancy between the FEM and experimental results on the hollow
nanoboxes. This indicates that a different material model, which includes mechanisms that lead
to hardening, is required to simulate the nanoboxes accurately. MD simulations are used to
capture this behavior. Thus, the fact that the FEM and experiments of the nanoboxes show
different results indicate that it is not the shape of the nanobox that gives rise to hardening.
Hardening at the structural level cannot be achieved without a corresponding material model that
includes hardening, which is in agreement with the reviewer's comments.

However, in line with the reviewer's comments, we have run additional FEM simulations in
order to account for the anisotropy of a single crystalline Ag nanobox. For an FCC metal, only 3
constants (c_{11} , c_{12} , and c_{44}) are necessary to fully populate the elastic stiffness matrix.⁶ The local
material coordinate system and global coordination are aligned so we are compressing in a
principal direction, which is in line with our experiments. The same loading conditions and
boundary conditions are used as previously. The resulting force-displacement curves closely
mimic that of an isotropic model, yielding around 230MPa at ~2% and a stress plateau (with
slight softening after that). This discussion has been added to the manuscript on page 17 line
359-364 and Supplementary Information.

Fig. 5. FEM simulation of a smooth nanobox using an anisotropic model. Cross-sectional view
 of **a** von Mises stress at ~1.5% strain, **b** radial displacement at ~18% strain, and **c** corresponding
 engineering stress-strain curve.

*Finally, MD simulations of the hollow nanocube are scaled down with respect to the experiment,*
 *using a cube side of 50 nm, and 3 different values of thickness: 7, 3, and 2 nm. They assume A_g*
 *as the single component of the cube. MD simulations are carried out at different compression*
 *rates, to test for strain rate effects, given that experiments are carried out at a much longer time*
 *scale. They give, for the sample with 1:7 ratio of wall thickness to cube length, a plastic yielding*
 *near 585 MPa and 4% strain, with strong hardening associated with densification when walls*
 *start to touch due to compression after 48% strain. Snapshots of the simulations indicate*
 *stacking faults, but there is no detailed dislocation analysis. For instance, the paper says “At*
 *53% strain, dislocations are found throughout the entirety of the sidewalls”. Fig. 6c shows*
 *evidence for stacking faults and some disordered regions, but most partial dislocations would*
 *have disappeared leaving behind only*
 *stacking faults. Full dislocations are unlikely to appear for such thin wall. Is there any twinning*
 *in the MD samples?*

Response: We apologize for the confusion and have updated the statement in the manuscript to:

“At 53% strain, defects left behind from dislocation slip and interactions including stacking
 faults and disordered regions are found throughout the entirety of the sidewalls”. The figure
 below shows a summary of the dislocations and stacking faults that appeared in our simulations.
 We do not observe full dislocations in any simulated sample. Twinning appears rarely in our
 simulations. Any twins are typically 3-4 atomic layers thick as shown in subplot (c) of figure
 below. We have expanded our discussion of deformation mechanisms as obtained from MD
 simulations in the manuscript on page 12, line 248-252.

 Fig. 6. Cross-sectional view of 1:7 smooth nanobox sample (ratio of wall thickness to length).
 Nanobox deformed to **a** 7% and **b** 23% strain showing nucleated stacking faults. Green atoms
 have FCC coordination and red atoms are stacking faults. **c,d,e,f** Enlarged regions marked by
 corresponding letter in **a,b**.

 *Regarding simulation of hollow nanostructures of fcc metals, there are two recent studies: one*

*focusing on elastic properties of Cu nanoshells [Yang 2018], and another including a study of*
*plastic properties for Pd nanoshells [Valencia2018]. Pd has much higher SFE, but still shows*
*mostly SFs and twins at high strain. A possible problem with the MD simulations presented in*
*the paper is the use of an interatomic potential not suited for plasticity under compression up to*
*large strains. Mordehai et al [Mordehai2018] showed that the plastic deformation of a solid*
*nanocube depends strongly on the potential used. There are many potentials for Ag*
*[OpenKim2019]. The EAM potential used in this work [Foiles 1986] was not fit to describe*
*properties related to dislocation plasticity, unlike more recent potentials like the one by Sheng*
*[Sheng 2011], used in [Sun 2014]. A recent study [Rassoulinejad-Mousavi 2018] shows that*
*elastic constants under pressure are not well described by this Ag*
*potential. Another recent study [Su 2019] shows that the [Sheng 2011] potential is the one which*
*compares better with ab-initio calculations of the generalized stacking fault energy surface,*
*which is one of the main factors in stacking fault nucleation and motion. The potential by Foiles*
*[Foiles 1986] gives a much lower value for the unstable and stable stacking faults. Therefore,*
*the observed strong hardening might be partly due to the (artificially) large number of stacking*
*faults due to this lower barrier.*

Response: We calculate the relaxed stable and unstable stacking fault energy of the Foiles
potential⁷ used here as 6.24 mJ/m² and 116 mJ/m² respectively. The same quantities for the
Sheng potential are 26 mJ/m² and 114 mJ/m².⁸ Thus, the energy barrier for creating the initial
stacking fault is similar between both potentials.

A study from Van Swygenhoven, et al. that investigated potentials and their likelihood of
nucleating full dislocations or partials with stacking faults showed that the ratio of the stable to
unstable stacking fault energy was a good metric to predict this likelihood⁹. Among their

potentials, only potentials with such a ratio >0.9 displayed full dislocations while all other
potentials (with maximum ratio of 0.7) displayed partials with extensive stacking faults. The
ratio of the Foiles potential and Sheng potential is 0.054 and 0.228 respectively, suggesting both
should display partials with stacking faults.

We repeated the MD simulations using the Sheng potential suggested by the reviewer. The figure
below shows a summary of generated stress-strain data and atomic snap-shots of deformation of
the nanobox using the Foiles potential and the Sheng potential. Quantitatively, the stress-strain
data is similar. There is an initially elastic regime which culminates in a peak at ~ 600 MPa.
There is then an intermediate regime with significant hardening before a final regime of
densification. The atomic deformation mechanisms are similar between the Foiles and Sheng
potentials, with initial deformation occurring by emission of partial dislocations from the free
surface, generating stacking faults through the thickness of the nanobox walls. Increasing
deformation leads to an increase in defect content with significant interaction between extended
stacking faults and subsequently emitted partial dislocations. This suggests that the Foiles
potential is able to capture the relevant deformation behaviour in these compression simulations
of Ag nanoboxes.

We thank the reviewer for bringing these other previous studies to our attention and have cited
them in the manuscript. We have also included the comparison between potentials and the above
discussion in the manuscript on page 19, line 392-400.

Fig. 7. MD simulations with the Foiles and Sheng potentials. Nanoboxes compressed to **a** 7%, **b**
 23%, **c** 53% strain using the Foiles potential. Nanoboxes compressed to **d** 7%, **e** 23%, **f**
 53% strain using the Sheng potential. **g** The stress-strain curves for the two potentials.

**Reviewer #3 comments:**

*This manuscript presents a combined experimental and computational study of plastic*
 *deformation and strain hardening during compression of hollow Au-Ag nanoboxes fabricated by*
 *galvanic replacement from perfect solid Ag nanocubes. The authors have tested metallic*
 *nanostructures of ~100 nm in size made with either smooth or rough surfaces. Using in situ*
 *SEM/TEM compression experiments, they observe that these nanoboxes exhibit strain-hardening*
 *behaviors exceeding that predicted theoretically for bulk Ag, more so when the surface of*

*nanoboxes is rough as opposed to smooth. FEA and MD simulations were used to study the*
*mechanism responsible for this behavior. It is concluded that the observed hardening effects*
*result from dislocation interaction with stacking-fault (SF) defects, rather than from structural*
*plastic buckling effects. This research is new and interesting, and the article is well-written and*
*well-presented. However, the main claim that the governing mechanism is*
*associated with dislocation-SF interactions should be better substantiated before this manuscript*
*could be accepted for publication in Nature Communications. In particular, the authors should*
*address the following major concerns:*

*1. The authors did not clearly address the internal microstructure differences between smooth*
*and rough nanoboxes, such as the presence of grain boundaries, which raises an important*
*concern that grain-boundary hardening should not have been omitted in the models.*

Response: The synthesis of nanoboxes, nanoshells (hollow nanocubes without holes at cube
corners) and nanocages using galvanic replacement is well established and covered in multiple
review articles (for example, Skrabalak et al., *Acc. Chem. Res.* 2008).¹⁰ In these studies, careful
microstructural analysis has been performed. For instance, high resolution TEM imaging and
TEM diffraction have shown that the resulting hollow structures are single crystalline.^{11,12} Our
nanoboxes are fabricated using identical synthetic protocols as in these previous studies, so they
are also expected to be single crystalline. We have added this point to the manuscript on page 6,
line 107-110, page 12, line 252-253. Additionally, as noted in response to reviewer #1 comments
(see above), from Fig. 1 (b), we see that the surface protrusions are also single crystalline and
with crystallographic orientation aligned with the base cube structure.

*2. The author should better address how dislocation-SF interactions could significantly*
*contribute to the overall hardening and how they are truly impacted by roughness effects.*
*Particularly, experimental and MD simulation evidence suggests that dislocation interactions*
*with pre-existing SFs are associated with very small transmission energy barriers (e.g. see Wang*
*et al. Nanoscale, 2019, 11, 12672). Therefore, the authors should propose a better model to*
*explain the observation that rough nanoboxes exhibit superior hardening. In its current version,*
*the manuscript remains unsettled on this question.*

Response: We have addressed the hardening in the rough nanoboxes in greater detail in response
to reviewer #1 comments (see above). In summary, from analysis of TEM images and MD
simulations of rough nanoboxes, we conclude that the rough protrusions act as stress
concentrations that lead to lower yield strength but higher hardening in the rough nanoboxes.
The reviewer refers to a paper entitled “Size-dependent dislocation-twin interactions” and
concerns a TEM and MD investigation into transmission of dislocations across a twin boundary.
As part of their analysis, the authors calculate the transmission energy across a twin boundary as
a function of the twin size. For typical {111} transmission, the authors show that the
transmission energy is independent of twin size at ~4 meV/atom for Au. We suppose that the
reviewer refers to the similarity between the twin and stacking fault in the small-size limit. The
energy barriers for slip transmission across a grain boundary in Cu are between 254.4-511.2
mJ/m².¹³ By comparison, the energy barriers for transmission of dislocations across a twin
boundary in Cu are reported to be between 199-232 mJ/m², which is similar to the lower bound
of grain boundary transmission¹³. Both of these interfaces are known to act as barriers to

dislocation motion and contribute to strengthening¹⁴. Although we are unable to find similar
reported quantities for FCC Ag, this suggests that in FCC Cu, a wide range of transmission
energies and interfaces can contribute to strengthening. We have included the preceding analysis
in the manuscript on page 12, line 242-248.

We note that in our simulations, gliding partial dislocations that encounter an existing stacking
fault tend to stop. Within the timescale of the simulation, transmission seldom occurs. An
increasing number of stacking faults therefore both limits the amount of plasticity generated by
individual dislocations and blocked dislocations create back-stresses requiring increasing applied
stress to nucleate subsequent dislocations. These mechanisms are shown in the manuscript in
Figure 6 and correlate with the stress-strain curves.

*3. FEA simulation results presented are not meaningful and should be further developed because*
*(1) they focused on only plastic buckling effects at strains < 20%, while structural effects in MD*
*simulations become only evident at strains > 40%, and (2) the elastic-plastic constitutive law*
*used for FEA is unclear. In fact, the authors should test their hypothesis by simulating*
*constitutive laws with different hardening exponents to study how it could affect the strain-strain*
*response of metallic nanoboxes. These results would be more insightful.*

Response: The value of the FEA simulations is discussed in the response to reviewer #1. To
summarize, the point of the FEA is to use the material model (perfectly elastic-plastic) that is
appropriate for a solid Ag nanocube (the parent structure for the nanobox) and test whether this
can result in hardening if it is applied to the nanobox geometry. The rationale behind this is to
test whether the nanobox behaves like the solid Ag nanocube with the interior and corners
removed, or whether new deformation mechanisms must be introduced. (1) In regard to

structural effects in FEA vs. MD simulation: The FEA simulation and MD simulation both show
bulging at the center of the side walls at ~20% strain (Figure 5b,6b). The value of the FEA
simulation is to show that this deformation mode occurs even in the absence of a material model
that includes hardening. (2) The constitutive law for FEA is the perfectly elastic-plastic material
model. This was chosen to mimic the behavior of solid Ag nanocubes in compression, which
showed no hardening after yielding.⁵

*4. The authors must clarify the following technical aspects in the text: (1) how stress was*
*computed in each method, especially in MD simulations, (2) how strain-hardening exponents*
*were measured experimentally, and if the last densification stage where stress increases sharply*
*was taken into account or not, and (3) if strain-hardening exponent was quantified in MD*
*simulations, and that such prediction should be compared to the experimental exponents.*

Response:

(1a) How stress was computed in MD simulations:

For the case of compression along the z-axis, the standard definition of engineering stress is
related to the atomic quantities that are computed during MD simulation according to:

$$\sigma_{zz} = \frac{F_z}{A_0} = \frac{\sum v s_{zz} / L_z}{L_{x0} L_{y0}}$$

where L_{x0} and L_{y0} are the initial cross-sectional dimensions of the nanobox, L_z is the height of
the nanobox, and v and s_{zz} are the atomic volume and atomic stresses, respectively. The product,
vs_{zz} , can be computed for each nanobox atom following standard definitions¹⁵ and then summed

over all nanobox atoms. L_z is taken as the separation distance between the two rigid plates and
L_{x0} and L_{y0} are computed at the beginning of the simulation using the extrema atomic positions.
This discussion is added on page 19, line 403-411 of the manuscript.

(1b) How stress was computed in experiments:

The stress was computed by dividing the measured force by the nominal (footprint) area of the
structures. For example, if the length of the nanobox was 140nm, the force was divided by 140 x
140 nm² to get the nominal engineering stress.

(2) How hardening exponents were measured experimentally:

The strain hardening exponents were derived by fitting an exponential relation of $\sigma = K\epsilon^n$ to
post yield experimental data for each individual test curve (this is discussed on page 8, line 143-
148 and page 17, line 344-349 of the manuscript). The values of n from all tests were then
averaged for the reported value of strain hardening exponent for the smooth and the rough
nanobox. The densification stage was not taken in the curve fitting as this sharp rise in stress is
due to touching sidewalls and not the material strain hardening effect. The fitting was limited up
to 50% strain to avoid the densification stage.

(3) If the strain hardening exponent was quantified in MD simulations:

Strain-hardening exponents were not computed for the generated MD stress-strain data primarily
due to the difficulty in numerical comparison between the experimental and computational
stress-strain data. For example, the high strain-rate of the simulated samples results in a much
larger yield stress (and consequently overall stress level) as compared to experiments. This is an
issue that occurs whenever MD simulations are compared to experiments. As a result, the MD

simulations are most valuable as a way to reveal deformation mechanisms and overall trends,
rather than to perfectly reproduce experimental data.

We have included the above expanded discussion of computation and experimental methods in
the Manuscript Methods section.

**References**

- 1. Pilkey, W. D. *Formulas for stress, strain, and structural matrices*. (John Wiley & Sons,
2005).
- 2. Greer, J. R. & De Hosson, J. T. M. Plasticity in small-sized metallic systems: Intrinsic
versus extrinsic size effect. *Prog. Mater. Sci.* **56**, 654–724 (2011).
- 3. Han, W.-Z. *et al.* From “Smaller is Stronger” to “Size-Independent Strength Plateau”:
Towards Measuring the Ideal Strength of Iron. *Adv. Mater.* **27**, 3385–3390 (2015).
- 4. Liu, Q., Wang, L. & Shen, S. Effect of surface roughness on elastic limit of silicon
nanowires. *Comput. Mater. Sci.* **101**, 267–274 (2015).
- 5. Kiani, M. T., Patil, R. P. & Gu, X. W. Dislocation surface nucleation in surfactant-
passivated metallic nanocubes. *MRS Commun.* 1–5 (2019).
- 6. Bower, A. F. & CRC Press. *Applied mechanics of solids*.
- 7. Foiles, S. M., Baskes, M. I. & Daw, M. S. Embedded-atom-method functions for the fcc
metals Cu, Ag, Au, Ni, Pd, Pt, and their alloys. *Phys. Rev. B* **33**, 7983–7991 (1986).
- 8. Sheng, H. W., Kramer, M. J., Cadien, A., Fujita, T. & Chen, M. W. Highly optimized

- embedded-atom-method potentials for fourteen fcc metals. *Phys. Rev. B* **83**, 134118
(2011).
- 9. Van Swygenhoven, H., Derlet, P. M. & Frøseth, A. G. Stacking fault energies and slip in
nanocrystalline metals. *Nat. Mater.* **3**, 399–403 (2004).
- 10. Skrabalak, S. E. *et al.* Gold Nanocages: Synthesis, Properties, and Applications. *Acc.*
*Chem. Res.* **41**, 1587–1595 (2008).
- 11. Hong, X., Wang, D., Cai, S., Rong, H. & Li, Y. Single-Crystalline Octahedral Au–Ag
Nanoframes. *J. Am. Chem. Soc.* **134**, 18165–18168 (2012).
- 12. Yugang Sun, Brian T. Mayers, A. & Xia, Y. Template-Engaged Replacement Reaction: A
One-Step Approach to the Large-Scale Synthesis of Metal Nanostructures with Hollow
Interiors. (2002). doi:10.1021/NL025531V
- 13. Sangid, M. D., Ezaz, T. & Sehitoglu, H. Energetics of residual dislocations associated
with slip–twin and slip–GBs interactions. *Mater. Sci. Eng. A* **542**, 21–30 (2012).
- 14. Lu, K., Lu, L. & Suresh, S. Strengthening Materials by Engineering Coherent Internal
Boundaries at the Nanoscale. *Science (80-.)*. **324**, 349–352 (2009).
- 15. Thompson, A. P., Plimpton, S. J. & Mattson, W. General formulation of pressure and
stress tensor for arbitrary many-body interaction potentials under periodic boundary
conditions. *J. Chem. Phys.* **131**, 154107 (2009).

Reviewers' Comments:

Reviewer #1:

None

Reviewer #3:

Remarks to the Author:

The authors have successfully addressed my first concern about grain-boundary effects, since it appears from the literature that the synthesized nanoboxes are single-crystalline.

The authors, however, have not provided convincing evidence that the origin of hardening directly resulted from dislocation-stacking-fault (SF) intersections, which is the primary hypothesis of this study. Given that the energy barrier from transmission of dislocations through SFs was found to be low in Au, all theoretical evidence suggests that the energy barrier for transmission in Ag should be very limited as well, because the SF energy of Ag is much lower than that of Au and Cu. In fact, further quantitative results from FEM analysis using a strain hardening constitutive law and MD simulations should be presented to convince me that the nanobox strain hardening does relate to dislocation-induced hardening from the sidewalls. In their current version, Figures 5, 6 and S1 are insufficient to substantiate this important claim.

Furthermore, the virial theorem used to compute the stress from their MD simulation is odd, as it is not clear if the authors have used the deformed or initial atomic volume in their equation. Using the initial atomic volume could yield inaccurate stress results when the simulation box is not periodic, large plastic deformation is applied and the model contains thin-walls. The authors should further clarify and verify their MD methodology.

While the manuscript quality is improving, I do not recommend its publication in Nature Communications yet without further evidence of the hardening mechanism at play.

Reviewer #4:

Remarks to the Author:

After carefully reviewing the answers to the reviewer's comments, I believe that the authors sufficiently address the main concerns and have expanded the technical robustness of the paper. However, I have two points for further clarification:

1. The FEM discussion did not appear to provide as much insight as the experimental tests or MD simulations. I think that using some language from the answers to the reviewers could clarify the FEM. For example, the authors state during their reply to reviewers 2 and 3 that : "The goal of the FEM simulations was actually not to reproduce the experimental results (this is left to the MD simulations, which are more appropriate for the size scale and complexity of the nanoboxes). Rather, the point of the FEM simulations is to answer the question, "What is the structural response for a material model that does not include strain hardening?" The reason that this is relevant to our system, is that the solid Ag nanocube (the parent structure from which the nanobox is formed) does not show strain hardening." Perhaps some version of this explanation could be incorporated into the text since it is not directly clear in the manuscript why the FEM presented is for a solid rather than a hollow system.

2. The Au:Ag ratios discussion is a bit confusing (pages 6&7). Are the authors implying that the Au:Ag ratios are 70/30 and 30/70? Those are two different alloys and perhaps the observed differences are highly impacted by the alloy composition. I think this section can be clearer and simply stating that the mechanical properties of Au and Ag are similar is not a sufficient explanation.

Response to Reviewer Comments (Second Round of Review)

**Hardening in Au-Ag Nanoboxes from Stacking Fault-Dislocation Interactions**

Radhika P. Patil^a, David Doan^a, Zachary H. Aitken^c, Shuai Chen^c, Mehrdad T. Kiani^b,
Christopher M. Barr^d, Khalid Hattar^d, Yong-Wei Zhang^c, X. Wendy Gu^{*a}

5 ^aDepartment of Mechanical Engineering, and ^bDepartment of Materials Science and Engineering,
Stanford University, Stanford, CA 94305, United States

7 ^cInstitute of High Performance Computing, A*STAR, 1 Fusionopolis Way, #16-16 Connexis,
138632 Singapore

9 ^dMaterials, Physical, and Chemical Sciences, Sandia National Laboratories, Albuquerque, NM,
87185, USA

**Reviewer #3 comments:**

*The authors have successfully addressed my first concern about grain-boundary effects, since it*
*appears from the literature that the synthesized nanoboxes are single-crystalline.*

*The authors, however, have not provided convincing evidence that the origin of hardening*
*directly resulted from dislocation-stacking-fault (SF) intersections, which is the primary*
*hypothesis of this study. Given that the energy barrier from transmission of dislocations through*
*SFs was found to be low in Au, all theoretical evidence suggests that the energy barrier for*
*transmission in Ag should be very limited as well, because the SF energy of Ag is much lower*

*than that of Au and Cu. In fact, further quantitative results from FEM analysis using a strain*
*hardening constitutive law and MD simulations should be presented to convince me that the*
*nanobox strain hardening does relate to dislocation-induced hardening from the sidewalls. In*
*their current version, Figures 5, 6 and S1 are insufficient to substantiate this important claim.*

Response: As explained in our previous reply to comments from reviewer#2 during the first
round of review (line 252), the goal of the FEM simulations was to analyze the structural
response (nanobox shape) without the influence of the material response. Adding a strain
hardening model in FEM would mean that we are attempting to reproduce the material response
using the strain hardening model, and would defeat the purpose of performing FEM.

In order to compare the transmission energy in the Ag potentials used here, we attempted to
reproduce the findings of the manuscript cited by the reviewer ¹ as reporting a low barrier for
dislocation transmission. We were unfortunately unable to reproduce the methodology because
the method is not provided in the cited paper. We do note several differences between the
structure used in that work and the microstructure observed here. In that paper, a leading partial
dislocation interacts with a single twinned zone (for a large twin region, this is in effect a single
planar fault). In our simulations, we observe bundles of stacking faults and small (2-3 atomic
layers) twinned regions. The result is that in our simulations, leading partial dislocations that are
able to “break through” a stacking fault or twin boundary quickly encounters another boundary.
The accumulated effect of these obstacles (even for low barriers) serves to create a hindrance to
slip that increases with a growing amount of dislocation content.

Additionally, although the energy barrier for transmission of dislocation is low in Au, there is
significant experimental evidence that stacking faults can accumulate in Au nanostructures. For

instance, stacking faults are found to pile up in deforming nanoporous Au nanowires which also
show hardening.² There are also reports of planar defects acting as barriers to dislocation motion
in Au. Bulge tests on single crystalline Au 105 nm thick films displayed strain hardening that
was attributed to the presence of pinned deformation twins³. In-situ TEM experiments show
stacking fault tetrahedra in Au acting as barriers to gliding dislocations, requiring significant
bowing of the dislocation before shearing through the stacking fault tetrahedral⁴.

There is also ample evidence from MD simulations on Au and other fcc metals that stacking fault
energies comparable to the ones used in our study are able to block dislocations. Previous MD
simulations subjected nanotwinned Au nanowires to tension and reported significant strain
hardening⁵. In these simulations, the authors employed an Au potential with stacking fault
energy of 43.4 mJ/m² and observed that the coherent twin boundaries served as strong barriers to
the glide of leading partial dislocations⁵. This phenomenon has also been observed in MD
simulations performed using a Cu potential with similar stacking fault and twin boundary energy
(44.42 mJ/m² and 22.27 mJ/m² respectively)⁶. These simulations showed a strong blocking
effect of twin boundaries to dislocations in nanotwinned Cu samples. We performed
compression simulations using two different Ag potentials with stacking fault energies of 6.24
mJ/m²⁷ and 26 mJ/m²⁸. In both sets of simulations, we observed stacking faults and twin
boundaries acting as impediments to gliding dislocations. Previous MD simulations considered
the interaction of a gliding partial dislocation with stacking fault tetrahedra in Cu using potentials
with similar stacking fault energies^{9,10}. In these studies, the authors employed Cu potentials with
stacking fault energies of 11.4 mJ/m²⁹ and 44.4 mJ/m²¹⁰ respectively. In both sets of
simulations, the stacking fault tetrahedra acted as strong barriers to dislocation motion, leading to
significant bowing of the leading partial dislocation.

We lastly note the difficulty in making any claims of transmissibility based solely on the planar
defect formation energy (i.e. the stacking fault energy in this case). For example, dislocation
emission and transmission through grain boundaries (GB) have been widely studied using MD
and there is a reported inverse relation between the transmission energy and GB energy ¹¹.

Fig. 1. Energy barrier for dislocation slip transfer across an Fe grain boundary

The above figure shows the energy barrier for dislocation slip transfer across a Fe GB as a
function of the GB energy and under two different hydrogen contents (figure from ¹¹). Clearly,
low energy GBs have a higher energy barrier for slip transmission. This can be understood in
terms of the stability of the interface. High energy boundaries tend to contain a large amount of
disorder and boundary free volume that serves as sites of dislocation nucleation or migration
sites in the case of dislocation transmission ^{12,13}. Low energy boundaries contain more coherent
sites, less boundary free volume and are thus more stable making transmission relatively more
difficult. The twin boundary (i.e. a highly-coherent $\Sigma 3$ boundary in the terminology of the
coherent site lattice) is a good example of a comparatively stable interface. These observations

and understanding are in contrast to the reviewer’s suggestion that a reduction in the stacking
fault planar boundary energy necessarily results in a reduction in the transmission barrier as to be
negligible for blocking dislocation motion.

In order to address the reviewer’s concern, we have included the following sentences on line 258
age 13 of the manuscript (note that super-scripts in the following text refer to manuscript
references).

“Previous MD simulations subjected nanotwinned Au nanowires to tension and reported
significant strain hardening that was attributed to coherent twin boundaries serving as strong
barriers to the glide of leading partial dislocations⁵. This phenomenon has also been observed in
MD simulations of nanotwinned Cu samples where twin boundaries were observed to have a
strong blocking effect on dislocations⁶. We note that both of these studies employed interatomic
potentials with similar low stacking fault energies, 43.4 mJ/m² for Au potential⁵ and 44.42 mJ/m²
Cu potential⁶ which compares to the stacking fault energies of 6.24 mJ/m²⁷ and 26 mJ/m²⁸ of
potentials used in this study.”

*Furthermore, the virial theorem used to compute the stress from their MD simulation is odd, as it*
*is not clear if the authors have used the deformed or initial atomic volume in their equation.*

*Using the initial atomic volume could yield inaccurate stress results when the simulation box is*
*not periodic, large plastic deformation is applied and the model contains thin-walls. The authors*
*should further clarify and verify their MD methodology.*

Response: The formula we have used to calculate stress of the nanobox in the MD simulations is:

$$\sigma_{zz} = \frac{F_z}{A_0} = \frac{\sum v s_{zz} / L_z}{L_{x0} L_{y0}}$$

In the equation, the numerator is computed using the standard definition of virial stress of the
instantaneous nanobox configuration. In analogy to experiments, this is the instantaneously
measured force. In order to match the experimental calculation of engineering stress, we divide
by the initial cross-sectional area of the nanobox sample.

In order to address the reviewer's concern, we have included the following sentences in the
manuscript on line 437 page 21.

"This form was chosen to most closely match the computation of engineering stress in
experiments. The numerator in Eq. 3 uses the standard calculation of virial stress using the
instantaneous sample configuration and is analogous to the instantaneous force as measured
during compression experiments. In order to match the experimental calculation of engineering
stress, we then divide by the initial cross-sectional area of the nanobox."

*While the manuscript quality is improving, I do not recommend its publication in Nature*
*Communications yet without further evidence of the hardening mechanism at play.*

**Reviewer #4 comments:**

*After carefully reviewing the answers to the reviewer's comments, I believe that the authors*
*sufficiently address the main concerns and have expanded the technical robustness of the paper.*

*However, I have two points for further clarification:*

*1. The FEM discussion did not appear to provide as much insight as the experimental tests or*
*MD simulations. I think that using some language from the answers to the reviewers could*

*clarify the FEM. For example, the authors state during their reply to reviewers 2 and 3 that :*
*“The goal of the FEM simulations was actually not to reproduce the experimental results (this is*
*left to the MD simulations, which are more appropriate for the size scale and complexity of the*
*nanoboxes). Rather, the point of the FEM simulations is to answer the question, “What is the*
*structural response for a material model that does not include strain hardening?” The reason*
*that this is relevant to our system, is that the solid Ag nanocube (the parent structure from which*
*the nanobox is formed) does not show strain hardening.” Perhaps some version of this*
*explanation could be incorporated into the text since it is not directly clear in the manuscript*
*why the FEM presented is for a solid rather than a hollow system.*

Response: The reviewer has misunderstood - the FEM simulation included in the manuscript is
of a hollow smooth nanobox, which was described on line 207 page 10 of the manuscript. FEM
of a solid silver nanocube was included in supplementary information (line 47 page 3) to check
that the solid cube results in the expected material response for an elasto-plastic material. In this
way, we know that we can trust the results for the hollow nanocube. As suggested by the
reviewer, we have clarified this issue and added an explanation of the purpose of the FEM
simulation on page 10 line 205 of the manuscript (note that super-scripts in the following text
refer to manuscript references):

“For further insight into the influence of structural architecture on deformation, a hollow smooth
nanobox is simulated using FEM analysis. The purpose of this FEM simulation is to determine
the structural response for a material that does not include strain hardening. This is relevant
because the solid Ag nanocube (the parent structure from which the nanobox is formed) does not
show strain hardening under compression.¹⁴ In this way, the FEM simulation reveals the
deformation of a hollow box, without consideration of material or size effects. Thus, an elastic

plastic material model that does not include strain hardening was used in the FEM simulation, to
mimic the material response of a Ag single crystal nanocube.¹⁴ The FEM model was then
constructed to have a similar geometry as the experimental nanobox, with smooth walls and a
hollow interior.”

*2. The Au:Ag ratios discussion is a bit confusing (pages 6&7). Are the authors implying that the*
*Au:Ag ratios are 70/30 and 30/70? Those are two different alloys and perhaps the observed*
*differences are highly impacted by the alloy composition. I think this section can be clearer and*
*simply stating that the mechanical properties of Au and Ag are similar is not a sufficient*
*explanation.*

Response: Yes, the estimated ratios of Au:Ag are 30:70 in the smooth nanobox samples and
70:30 in the rough nanobox samples. The discussion of the elemental ratios is intended to notify
readers of this difference in compositions for the two samples. By stating that the mechanical
behavior of Au and Ag are similar, we mean that the Au-Ag interface is not expected to
significantly impact dislocation and stacking fault transmission. This has been observed in our
previous study of Au-Ag core shell nanocubes,¹⁵ in which dislocations were able to cross the Au-
Ag interface, such that the stress-strain response of the Au-Ag nanocubes was similar to that of
solid Ag nanocubes.¹⁴ This behavior is not unexpected, considering the interactions of
dislocations with bimetallic interfaces that are used to design nanolaminates and precipitate
hardened alloys.¹⁶⁻¹⁸ Au and Ag form a coherent interface without significant lattice mismatch.
This type of interface does not provide significant strengthening, or obstacles to dislocations, and

will not block the passage of dislocations. This issue has been clarified in the manuscript on line
124 on page 7:

“This difference in composition is not expected to cause a significant difference in the properties
of the smooth and rough nanoboxes. The Ag and Au interface does not impede the passage of
dislocations or stacking faults, such that the difference in composition should not affect the
deformation mechanism for the nanoboxes.¹⁵”

**References**

- 1. Wang, J., Cao, G., Zhang, Z. & Sansoz, F. Size-dependent dislocation–twin interactions.
*Nanoscale* **11**, 12672–12679 (2019).
- 2. Dou, R. & Derby, B. Deformation mechanisms in gold nanowires and nanoporous gold.
*Philos. Mag.* **91**, 1070–1083 (2011).
- 3. Catlin, A. & Walker, W. P. Mechanical Properties of Thin Single-Crystal Gold Films. *J.*
*Appl. Phys.* **31**, 2135–2139 (1960).
- 4. Matsukawa, Y. & Zinkle, S. J. Dynamic observation of the collapse process of a stacking
fault tetrahedron by moving dislocations. *J. Nucl. Mater.* **329–333**, 919–923 (2004).
- 5. Deng, C. & Sansoz, F. Enabling Ultrahigh Plastic Flow and Work Hardening in Twinned
Gold Nanowires. *Nano Lett.* **9**, 1517–1522 (2009).
- 6. Wu, Z. X., Zhang, Y. W. & Srolovitz, D. J. Dislocation–twin interaction mechanisms for
ultrahigh strength and ductility in nanotwinned metals. *Acta Mater.* **57**, 4508–4518
(2009).

- 7. Foiles, S. M., Baskes, M. I. & Daw, M. S. Embedded-atom-method functions for the fcc
metals Cu, Ag, Au, Ni, Pd, Pt, and their alloys. *Phys. Rev. B* **33**, 7983–7991 (1986).
- 8. Sheng, H. W., Kramer, M. J., Cadien, A., Fujita, T. & Chen, M. W. Highly optimized
embedded-atom-method potentials for fourteen fcc metals. *Phys. Rev. B* **83**, 134118
(2011).
- 9. Wirth, B. D., Bulatov, V. V. & de la Rubia, T. D. Dislocation-Stacking Fault Tetrahedron
Interactions in Cu. *J. Eng. Mater. Technol.* **124**, 329–334 (2002).
- 10. Marian, J., Martínez, E., Lee, H.-J. & Wirth, B. D. Micro/meso-scale computational study
of dislocation-stacking-fault tetrahedron interactions in copper. *J. Mater. Res.* **24**, 3628–
3635 (2009).
- 11. Adlakha, I. & Solanki, K. N. Critical assessment of hydrogen effects on the slip
transmission across grain boundaries in α -Fe. *Proc. R. Soc. A Math. Phys. Eng. Sci.* **472**,
20150617 (2016).
- 12. Tschopp, M. a., Tucker, G. J. & McDowell, D. L. Structure and free volume of $\langle 110 \rangle$
symmetric tilt grain boundaries with the E structural unit. *Acta Mater.* **55**, 3959–3969
(2007).
- 13. Tucker, G. J., Tschopp, M. A. & McDowell, D. L. Evolution of structure and free volume
in symmetric tilt grain boundaries during dislocation nucleation. *Acta Mater.* **58**, 6464–
6473 (2010).
- 14. Kiani, M. T., Patil, R. P. & Gu, X. W. Dislocation surface nucleation in surfactant-
passivated metallic nanocubes. *MRS Commun.* 1–5 (2019).

- 15. Kiani, M. T., Wang, Y., Bertin, N., Cai, W. & Gu, X. W. Strengthening Mechanism of a
Single Precipitate in a Metallic Nanocube. *Nano Lett.* **19**, 255–260 (2019).
- 16. Beyerlein, I. J., Demkowicz, M. J., Misra, A. & Uberuaga, B. P. Defect-interface
interactions. *Progress in Materials Science* **74**, 125–210 (2015).
- 17. Beyerlein, I. J. *et al.* Structure-property-functionality of bimetal interfaces. *JOM* **64**,
1192–1207 (2012).
- 18. Nembach, E. *Particle strengthening of metals and alloys*. (Wiley, 1997).

Reviewers' Comments:

Reviewer #4:

Remarks to the Author:

The authors have fully addressed my concerns and modified the manuscript accordingly. The paper should be accepted.